# Physical constraints and functional plasticity of cellulases

Jeppe Kari [1,6], Gustavo A. Molina [1,6], Kay S. Schaller[1], Corinna Schiano-di-Cola [1], Stefan J. Christensen[1], Silke F. Badino[1], Trine H. Sørensen[2], Nanna S. Røjel [3], Malene B. Keller [4], Nanna Rolsted Sørensen[3], Bartlomiej Kolaczkowski[3], Johan P. Olsen [2], Kristian B. R. M. Krogh[2], Kenneth Jensen[2], Ana M. Cavaleiro[2], Günther H. J. Peters [5], Nikolaj Spodsberg [2], Kim Borch[2] & Peter Westh [1✉]

Enzyme reactions, both in Nature and technical applications, commonly occur at the interface of immiscible phases. Nevertheless, stringent descriptions of interfacial enzyme catalysis remain sparse, and this is partly due to a shortage of coherent experimental data to guide and assess such work. In this work, we produced and kinetically characterized 83 cellulases, which revealed a conspicuous linear free energy relationship (LFER) between the substrate binding strength and the activation barrier. The scaling occurred despite the investigated enzymes being structurally and mechanistically diverse. We suggest that the scaling reflects basic physical restrictions of the hydrolytic process and that evolutionary selection has condensed cellulase phenotypes near the line. One consequence of the LFER is that the activity of a cellulase can be estimated from its substrate binding strength, irrespectively of structural and mechanistic details, and this appears promising for in silico selection and design within this industrially important group of enzymes.

[1] Department of Biotechnology and Biomedicine, Technical University of Denmark, Kongens Lyngby, Denmark. [2] Novozymes A/S, Bagsværd, Denmark. [3] Department of Science and Environment, Roskilde University, Universitetsvej 1, Roskilde, Denmark. [4] Department of Geosciences and Natural Resource Management, University of Copenhagen, Frederiksberg C, Denmark. [5] Department of Chemistry, Technical University of Denmark, Kongens Lyngby, Denmark. [6] These authors contributed equally: Jeppe Kari, Gustavo A. Molina. ✉email: petwe@dtu.dk

Enzyme reactions at interfaces are common both in Nature and industry[1]. About half of the enzymes in the living cell work at a membrane surface[2] and many technical enzyme applications involve catalysis at the solid-liquid interface[3]. Examples of the latter include the use of immobilized enzymes in protein arrays or biosensors[4], but more commonly, the activity of soluble enzymes on insoluble substrates such as polysaccharides, lipids, precipitated proteins[5] or more recently plastic[6]. Studies of heterogeneous enzyme reactions have shown that substrate specificity[7], turnover number[8], and enzyme-substrate binding affinity[9] can be significantly altered at an interface compared to analogous reactions in the bulk. Nevertheless, the kinetics of interfacial reactions is typically disregarded or fleetingly treated in textbooks[10-14], and this state of affairs is quite different from conventional (non-biochemical) catalysis, where homogeneous and heterogeneous reactions are treated in parallel. Although insightful models and concepts of interfacial enzyme kinetics have been suggested[15-17], no generally applied kinetic approach or rate equation currently exist. Neither is it clear whether progress in this field should be based on adaptation of conventional enzyme kinetic theory, or modifications of concepts and principles taken from inorganic heterogeneous catalysis.

Here we investigated heterogeneous enzyme catalysis using cellulases as a paradigm. These enzymes catalyze the hydrolysis of the β-1,4 glycosidic bond that links glucopyranose units in (insoluble) cellulose and constitute a generic and experimentally convenient example of interfacial enzymes. In addition, cellulases are of direct industrial interest since enzymatic conversion of lignocellulosic biomass into fermentable sugars (known as saccharification) is expected to play a key role in the upcoming biorefineries that produce fuels, chemicals, and materials from sustainable feedstocks[18-20]. We focused on fungal cellulases, which are commonly applied in industrial enzyme cocktails[21], and investigated enzymes from Glycoside Hydrolase (GH) family 5, 6, 7, 12, and 45[22]. Specifically, we produced and biochemically characterized 83 enzymes using insoluble cellulose as substrate. The characterized cellulases included both, wild types and variants, and represented a wide range of structural and functional differences (see Fig. 1). Nevertheless, the kinetic data showed a clear common trait as we found a conspicuous scaling between the apparent Michaelis–Menten (MM) constant ($K_M$) and the maximal turnover ($k_{cat}$) across the entire group of cellulases. The scaling could be expressed as a so-called linear free energy relationship (LFER), and we used this to discuss functional plasticity and physical constraints for the enzymatic conversion of cellulose. We argue that the LFER for cellulases may facilitate both

mechanistic and evolutionary studies, and act as guidance in future attempts to select or design improved technical enzymes. Moreover, the observed LFER is reminiscent of the behavior found for some well-described inorganic heterogeneous catalysts, and this may help to establish better theoretical frameworks for interfacial enzyme reactions.

## Results

**Enzyme production.** The investigated enzymes were selected from five GH families as illustrated in Fig. 1 and Table 1. These families (GH5, GH6, GH7, GH12, and GH45) cover essentially all major fungal cellulases[21] and hence represent a wide range of structures and mechanisms. This included enzymes with or without a carbohydrate-binding module (CBM), enzymes using an inverting or retaining mechanism, enzymes that attack the cellulose chain internally (endoglucanases, EGs) or at a chain end (cellobiohydrolases, CBHs) and enzymes with different degrees of processivity. In addition to the wild types, a library of cellulase variants was made with the intention of changing the enzyme-substrate binding strength. This library included variants with mutations in the CBM, linker, and catalytic domain, as well as variants, where the CBM and linker were added, removed or swapped. A full list of the enzymes characterized here, can be found in supplementary Table 4 in the supplementary information (SI).

**Kinetic analysis.** All enzymes were characterized by MM kinetics using microcrystalline cellulose (Avicel PH-101) as substrate. Quasi-steady-state rates ($v_{ss}$) were measured at a constant, low enzyme concentration ($E_0$) and different substrate loads ($S_0$), and analyzed by the MM-equation (Eq. 1) using non-linear regression. The resulting kinetic parameters ($K_M$ and $k_{cat}$) are listed in supplementary Table 4. Previous studies have identified practical procedures for measuring the quasi-steady-state rate for this type of system[23] and shown that Eq. 1 is valid and applicable even though the substrate is solid and specified by its mass load ($S_0$) in units of g/L[24,25]. The derived rates were based on soluble products only and control experiments (supplementary Table 3) showed that this was a good descriptor of the overall activity even for EGs.

$$v_{ss} = \frac{E_0 k_{cat} S_0}{S_0 + K_M} \qquad (1)$$

In Fig. 2, the natural logarithm of the derived kinetic parameters ($K_M$ and $k_{cat}$) are plotted against each other for all

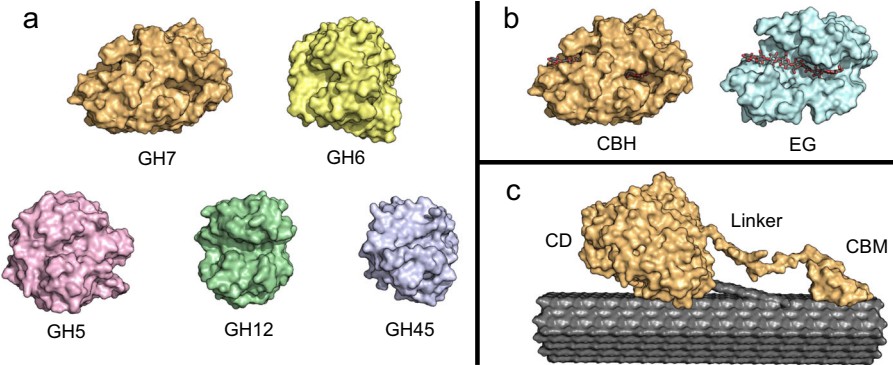

**Fig. 1 Structural representation of the different classes of cellulases characterized in this study. a** Surface representation of the six different glycoside hydrolase (GH) families (exemplified by the PDB ID: 4C4C[53] (GH7), 1QK2[55] (GH6), 1H8V[56] (GH12), 4ENG[58] (GH45), 3QR3[57] (GH5)). **b** Structure of two GH7 cellulases with different modes of action in complex with cellononaose. A cellobiohydrolase (CBH) with a tunnel-shaped catalytic domain (PDB: 4C4C) and an endoglucanase (EG) with an open catalytic cleft (PDB: 1EG1[54]). **c** Illustration of a GH7 CBH in complex with a cellulose fiber. The enzyme is modular with a catalytic domain (CD) and a carbohydrate-binding module (CBM) connected by a flexible linker. All structures were visualized using PyMOL[71].

**Table 1 Fungal cellulases characterized in this study.**

| GH family | Structural fold | Catalytic mechanism | Mode of action | EC number | Number of enzymes | |
|---|---|---|---|---|---|---|
| | | | | | Wild-types | Variants |
| 7 | β-jelly roll | Retaining | CBH | 3.2.1.176 | 11 | 21 |
| | | | EG | 3.2.1.4 | 3 | 6 |
| 6 | α/β barrel | Inverting | CBH | 3.2.1.91 | 5 | 16 |
| 5 | (β/α)$_8$ | Retaining | EG | 3.2.1.4 | 8 | 0 |
| 12 | β-jelly roll | Retaining | EG | 3.2.1.4 | 3 | 4 |
| 45 | β barrel | Inverting | EG | 3.2.1.4 | 6 | 0 |
| Total | | | | | | 83 |

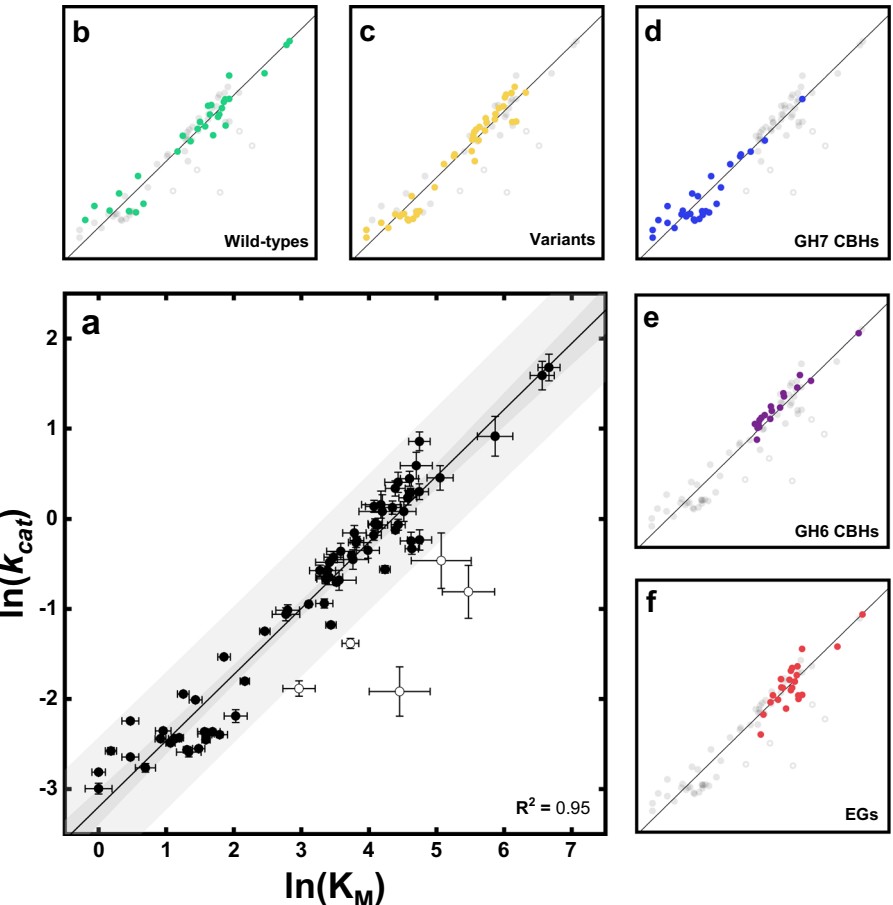

**Fig. 2 Correlation plot of ln(K$_M$) and ln(k$_{cat}$).** Correlation plot for all investigated enzymes (**a**). The smaller panels highlight data for different classes of enzymes. These are wild type cellulases (**b**), variants (**c**), cellobiohydrolases from GH7 (**d**), cellobiohydrolases from GH6 (**e**), and endoglucanases from family GH7, GH12, and GH45 (**f**). The solid line in all plots derives from linear regression to the experimental data in the main panel (**a**) excluding the outliers (open symbols) identified as explained in the main text. Bands shown in panel (**a**) are 95% confidence band (dark gray) and 95% prediction band (light gray) of the linear regression. Error bars in (**a**) represent standard deviations from MM-fit (Eq. 2) to triplicates.

investigated enzymes to illustrate the power-law correlation between the kinetic parameters. From the main panel (Fig. 2a), it appeared that most enzymes clustered in a narrow lane around the diagonal. Some enzymes were located below the diagonal, but we did not find any above. To assess whether the experimental points in Fig. 2 correlated with structural or functional properties of the studied enzymes, we highlighted specific sub-groups in the dataset in five separate subplots (Panels b–f, Fig. 2). Linear regression showed that the slope in Fig. 2 was 0.74 ± 0.02. Regression outliers were identified based on studentized residual analysis using a conservative cutoff of ±2.5σ. The outliers were omitted from the regression analysis and identified by open

symbols in Fig. 2. A list of kinetic parameters for all investigated enzymes can be found in supplementary Table 4.

**Computational analysis**. The strong correlation between ln(K$_M$) and ln(k$_{cat}$) shown in Fig. 2 is attractive from a computational point of view. If the apparent MM constant (K$_M$) can be interpreted as a descriptor for the enzyme-substrate binding affinity it may open up for prediction of catalytic rates based solely on computed binding free energies. To test this hypothesis we computed cellulose binding strengths for a subset of nine enzymes from Fig. 2, using molecular dynamics (MD)

simulations with umbrella sampling along the binding path (See further details in supplementary Fig. 8). We selected enzymes that spanned a wide range of $K_M$ values and represented all structural and functional classes listed in Fig. 1. For modular cellulases, the contribution of the CBM to binding energy $\Delta G°_B$ was computed separately. To compare with experiments we used $k_{cat}$ and $K_M$, to estimate changes in respectively transition-state free energy ($\Delta\Delta G^{\ddagger}$) and standard free energy of ligand binding ($\Delta\Delta G°_B$) following well-established principles[26–28]. Specifically, we used the equations

$$\Delta\Delta G_B^o = RT\ln\left(\frac{K_M}{K_{M,ref}}\right) \quad (2)$$

$$\Delta\Delta G^{\ddagger} = -RT\ln\left(\frac{k_{cat}}{k_{cat,ref}}\right) \quad (3)$$

which introduce a reference enzyme with the kinetic parameters $K_{M,ref}$ and $k_{cat,ref}$. Hence, the calculated free energies are energy changes relative to the selected reference. This approach alleviates ambiguities regarding standard states (Eq. 2) and pre-exponential factors (Eq. 3). We used the GH6 cellobiohydrolase from *Trichoderma reseei* (TrCel6A) as our reference enzyme and it follows that this enzyme will have $\Delta\Delta G^{\ddagger} = \Delta\Delta G°_B = 0$.

The validity of Eq. 2 is dependent on whether $K_M$ can be interpreted as a descriptor of the enzyme-substrate affinity. The comparison in Fig. 3a showed that despite the diversity of the analyzed cellulases, computed changes in binding affinity, $\Delta\Delta G_{B,MD}$, scaled reasonably well with the experimental values, $\Delta\Delta G_{B,exp}$, derived from Eq. 2. This supports the validity of Eq. 2 for this system and the idea of using computed ligand-binding energies to predict catalytic rates. Figure 3b illustrate the scaling between $\Delta\Delta G_{B,MD}$ and $\Delta\Delta G^{\ddagger}_{exp}$.

## Discussion

In this study, we produced and kinetically characterized 83 enzymes covering essentially all classes of fungal cellulases (Table 1 and Fig. 1). We used the same expression host, to ensure the enzymes were exposed to the same apparatus of post-translational modifications. Moreover, kinetic characterizations were based on the same substrate, experimental conditions, and principles of analysis. This provided a robust basis for comparative analyses of interfacial enzymes in general and cellulases in particular. Indeed, the breadth of the dataset allowed us to identify a striking correlation between $\ln(K_M)$ and $\ln(k_{cat})$ and in the following we discuss the origin and corollaries of this observation.

**Enzyme fitness and physical constraints**. Figure 2 may be seen as a fitness landscape for cellulases attacking their native insoluble substrate, and it appears that most enzymes accumulated around the diagonal. The diagonal defines a continuum ranging from enzymes with weak substrate interactions and rapid turnover (high $K_M$ and $k_{cat}$), to enzymes with stronger interactions, but slower turnover (low $K_M$ and $k_{cat}$). The tendency to accumulate along the diagonal was observed for all types of cellulases (refer to Table 1 and Fig. 1), and hence does not seem to rely on specific structural or mechanistic properties. Rather, it appears that the maximal turnover can be expressed solely by one descriptor, namely $K_M$. The area below the diagonal in Fig. 2 represents a region where the enzymes have a low specificity constant (i.e., low $k_{cat}/K_M$), and this seems to signify inefficient catalysis. We found some enzymes in this range, including some wild type enzymes and variants with replacements of key amino acid residues. We suggest that this southeastern region of the fitness landscape represents enzymes that have been either catalytically impaired by our engineering, are structurally unstable under the selected conditions, or have other primary substrate preference than cellulose.

On the other hand, the region above the diagonal in Fig. 2, specifies enzymes, which have a high specificity constant on this substrate. This clearly appears functionally advantageous, but we did not find any cellulases in this northwestern region. We suggest that this absence is the result of basic physical restrictions of the cellulolytic process. It follows that the accumulation of data points in a narrow lane in Fig. 2 may be seen as a balance between

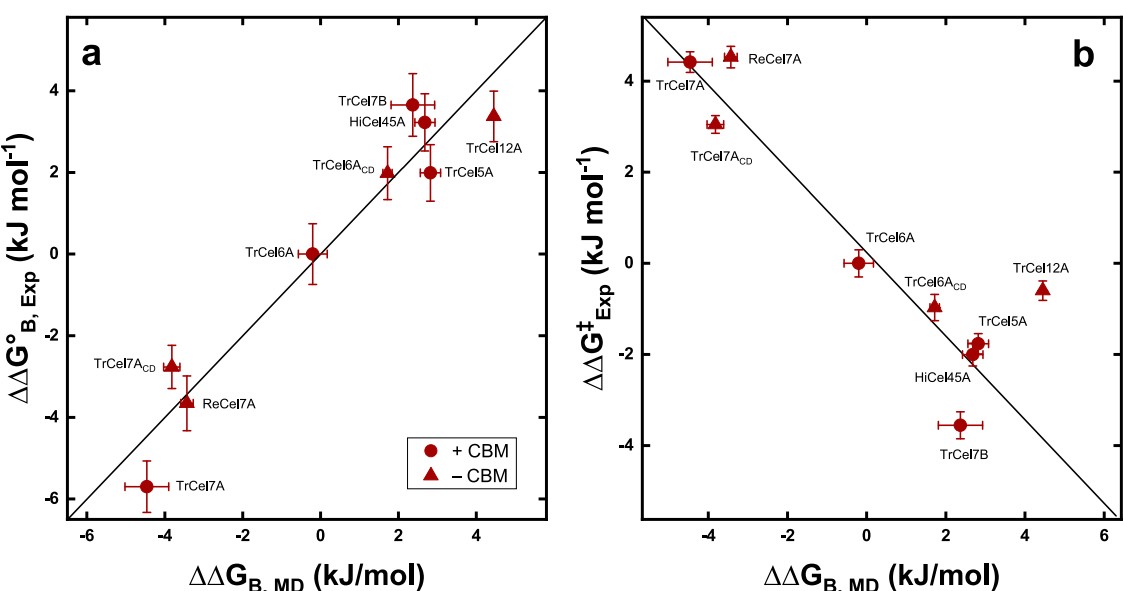

**Fig. 3 Correlation of computed and experimental free energies for nine selected cellulases. a** Changes in computed free energies of binding ($\Delta\Delta G_{B,MD}$) and experimental changes in binding free energy ($\Delta\Delta G_{B,exp}$). **b** Correlation of $\Delta\Delta G_{B,MD}$ and experimental changes in activiation free energy ($\Delta\Delta G^{\ddagger}_{exp}$). Experimental free energies were calculated using Eqs. 2 and 3. The kinetic parameters ($K_M$ and $k_{cat}$) of the nine cellulases can be found in supplementary Table 4. The selected cellulases covered a wide range of kinetic parameters shown in Fig. 2 and encompassed all main structural and functional traits specified in Fig. 1. Standard deviations of the experimental free energies and computed free energies are shown as error bars.

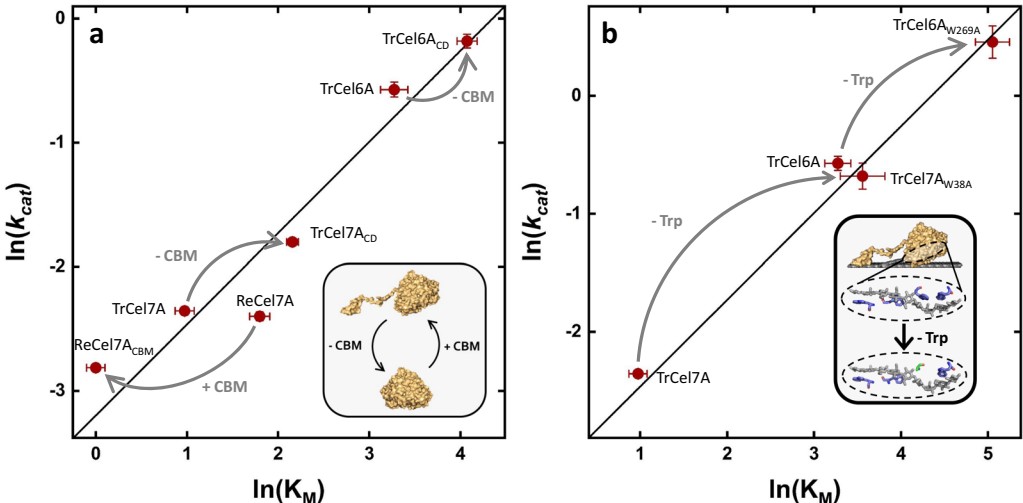

**Fig. 4 Illustration of the effect of the non-catalytic CBM (a) and tryptophan residues in the catalytic domain of cellobiohydrolases from GH6 and GH7 (b). a** Correlation plot of $\ln(K_M)$ and $\ln(k_{cat})$ for three wild-type CBHs and three variants, where the CBM was either removed (−CBM) from the wild-type (TrCel7A → TrCel7A$_{CD}$, TrCel6A → TrCel6A$_{CD}$) or added (+CBM) to the wild type (ReCel7A → ReCel7A$_{CBM}$). **b** Analogous correlation plot for replacements of conserved tryptophan residues by alanine in the catalytic domain of TrCel7A (TrCel7A → TrCel7A$_{W38A}$) or TrCel6A (TrCel6A → TrCel6A$_{W269A}$). The solid line shown in both plots is the same as in Fig. 2. It appears that changes in $K_M$ and $k_{cat}$ tend to compensate so that all enzymes remain close to the diagonal. Inserts are illustrations to guide the reader about the structural changes in the variants. Error bars represent standard deviations from MM-fit (Eq. 2) to triplicates.

evolutionary selection, which drives the kinetic parameters toward the northwest, and physical constraints, which prevents this development beyond the boundary defined by the line in Fig. 2.

The engineered variants in Fig. 2b represent a range of replacements and deletions at different positions (see supplementary Table 4), which were designed with the overall purpose of altering ligand-binding strength. In a few cases, the mutations shifted the variants into the southeastern "wasteland" of the fitness landscape, but most remained on the diagonal. The tendency to stay on the line did not reflect that the variants had unaltered kinetic parameters. Rather, changes in $K_M$ and $k_{cat}$ tended to compensate. Some examples of this are highlighted in Fig. 4, and it appears that both point mutations, and extensive changes in the amino acid sequence, readily moved kinetic parameters up or down the diagonal, but rarely sent them off the line. Interestingly, the vast majority of the variants moved up the line compared to their respective wild type, and only in cases where a CBM was added to a CBM-less wild type (Fig. 4a) did the variant move down the line toward lower $K_M$ and $k_{cat}$ values. This indicates that the wild-type enzymes have evolved to have high affinity for the substrate rather than high turnover. Nonetheless, the differences in affinities across GH families may be important in Nature where cellulose is degraded by cellulases from multiple GH families.

**Origin of physical constraint.** Correlations between binding and activation free energies are well-known in both organic and inorganic catalysis[29], but have only been sporadically used for (homogenous) enzyme reactions[30,31]. A LFER exists, if the binding free energy, $\Delta G^\circ_B$, scales linearly with the free energy of activation, $\Delta G^\ddagger$. This is tantamount to proportionality between the changes in these two free energies, and we may write

$$\Delta\Delta G^\ddagger = \Phi\Delta\Delta G^o_B \qquad (4)$$

where $\Phi$ is a scaling constant that convert changes in the binding free energy ($\Delta\Delta G^o_B$) to changes in activation free energy ($\Delta\Delta G^\ddagger$). The correlation shown in Fig. 2 may be interpreted as an LFER if

the $K_M$ values can be interpreted as a dissociation constant for the enzyme-substrate complex. In general one has to be cautious when using the (apparent) $K_M$ value as affinity descriptor for complex enzyme reactions such as the one studied here. However, such interpretation of $K_M$ has been successfully used earlier[26–28] and it is also in line with the MD results (Fig. 3a) that showed good correlations between computed ligand-binding energies and experimental binding energies calculated using Eq. 2. The validity of $K_M$ as a descriptor of the enzyme-substrate affinity of the investigated enzymes is further discussed in the SI (see supplementary note 1 and 2).

Using Eqs. 2 and 3 we calculated $\Delta\Delta G^\circ_B$ and $\Delta\Delta G^\ddagger$ and found that the two free energies correlated with a slope of $\Phi = -0.74 \pm 0.02$ (see supplementary Fig. 9). This is the same slope as found for the line in Fig. 2 but with opposite sign due to the minus in Eq. 3 (e.g., low activation energies gives high $k_{cat}$ values). The scaling constant, $\Phi$ in Eq. 4 provides some information about the nature of the transition state (TS), and this idea has been used, for example, to elucidate the TS of protein folding[32]. As proposed by Warshel[27], the $\Phi$-value also provides a means to classify effects of mutations on enzyme function. If, for example, both the enzyme-substrate complex and the TS in a variant are stabilized to the same extent (so-called uniform binding, see Figs. 5b–1) $\Phi$ would be 0 since the activation energy would remain unchanged (i.e., and $\Delta\Delta G^\ddagger = 0$). Another illustrative case is when changes in interactions only manifest themselves in the TS (so-called TS-stabilization, Figs. 5b–2). This results in $\Phi \rightarrow \propto$ since the activation energy can be changed independently of the binding energy. Finally, if mutations only act to stabilize the ground state complex (GS stabilization, Figs. 5b–3), $\Delta\Delta G^\ddagger$ will change commensurate with $\Delta\Delta G^\circ_B$, and $\Phi = -1$.

This interpretation of $\Phi$-values was developed to classify mutants that were closely related in structure, but in the current context it may elucidate differences across cellulases (wild types and mutants) with widely different structures and mechanisms. We found $\Phi = -0.74 \pm 0.02$ (see supplementary Fig. 9), and it follows that kinetic differences among the investigated cellulases can be mostly ascribed to differences in the degree of GS stabilization. This has the noticeable consequence that the free

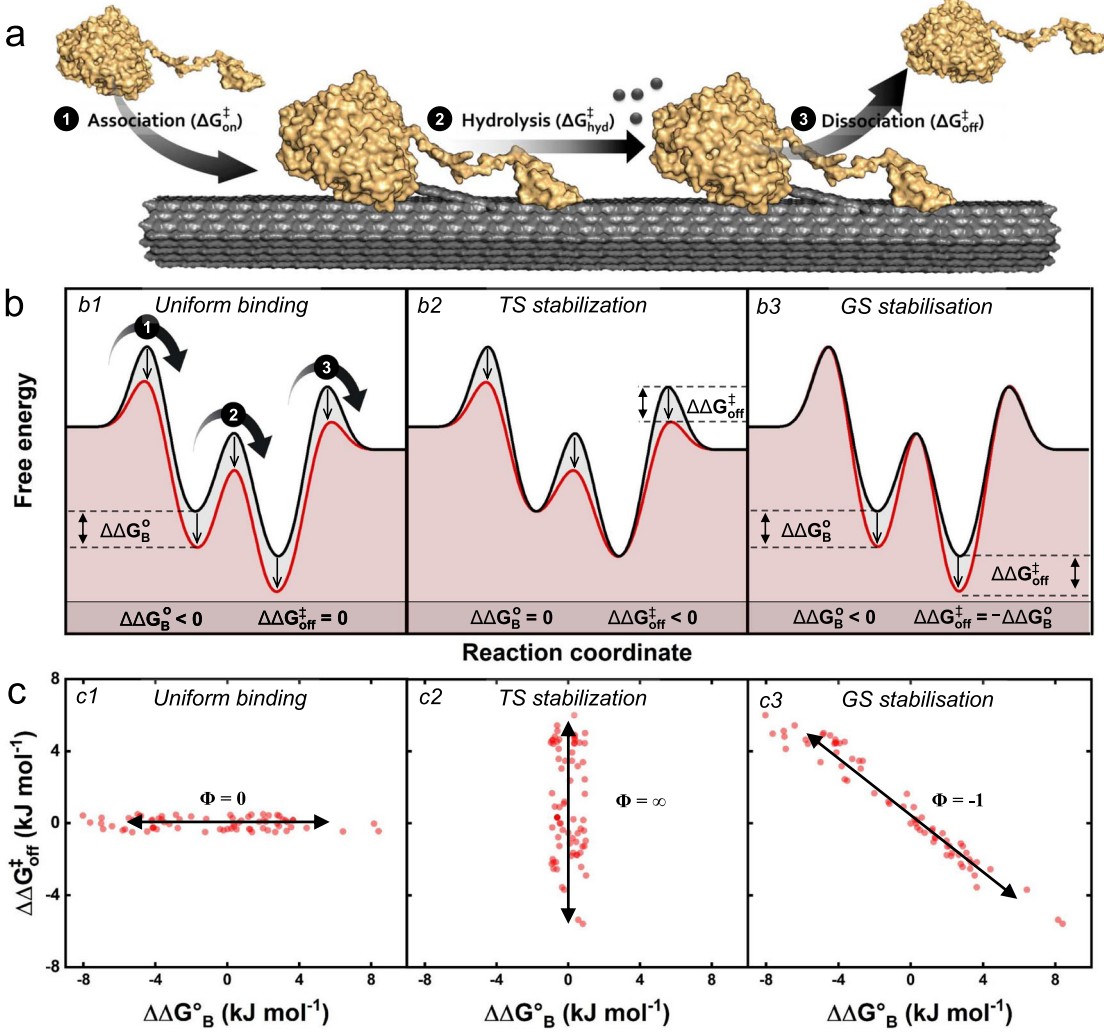

**Fig. 5 Structural and energetic interpretation of a simplified reaction scheme for the enzyme-catalyzed hydrolysis of cellulose. a** Simplified reaction scheme for a cellulase (yellow) hydrolyzing insoluble cellulose (gray). The cartoon provides a structural interpretation of the three steps in the overall reaction; (1) association, (2) hydrolysis, and (3) dissociation. **b** Schematic energy-diagrams for a wild type (black curve) and three conceptually different variants (red curves). **c** Expected scaling plots for a group of variants that behave according to the three different energy-diagrams shown in (**b**). If the energy of the variant differs from the wild type by the same amount in both transition state (TS) and ground state (GS), we have so-called uniform binding and $\Phi = 0$ (panel b1 and c1). The parallel shift in energies for uniform binding implies that the same interactions occur in GS and TS. If, on the other hand, a mutation only lowers the TS energy, known as TS-stabilization, this leads to a vertical line in the scaling plot (panel b2 and c2). Finally, in GS-stabilization (panels B3 and c3), only the GS energy changes, while the TS remains fixed. In this case, $\Phi = -1$, and this is close to the experimental observation (see supplementary Fig. 9).

energy of the (rate-limiting) TS is quite similar for all tested enzymes, and that the main kinetic diversity lies in different affinities for the substrate. This is illustrated in Figs. 5b–3, which shows that tighter binding to the substrate (red trace) unavoidably leads to a higher activation energy if the TS is (almost) fixed. Experimental studies have suggested that the rate-limiting step for some cellulases is slow dissociation[33–37]. Since weaker binding is associated with a lower activation barrier for dissociation (Figs. 5b–3), a dissociation limited mechanism would explain the inverse correlation of binding strength and maximal turnover. Based on these considerations it is tempting to suggest that weak ligand-binding is a functional advantage since it invariably increases $k_{cat}$. However, mutational studies suggest that weak binding is not necessarily advantageous for the efficacy of GHs attacking solid carbohydrates[38,39]. The characterized variants support this interpretation, since most of the variants moved up the line in Fig. 2 compared to the respective wild type,

indicating that the wild types were optimized for high affinity. Strong ligand binding may be needed in order for the enzyme to transfer a cellulose chain from the cellulose surface, where it is strongly bound[40,41], to the binding cleft (see cartoon in Fig. 5a). Hence, strong ligand binding appears to benefit catalysis by promoting ligand transfer[42], but it is inevitably associated with a slow turnover of an off-rate controlled reaction, as illustrated in Figs. 5b–3. We suggest that the LFER between the binding energy and activation energy, is a direct consequence of the overall reaction being controlled by the on-off kinetics of the cellulases (see supplementary Fig. 6). The existence of LFERs for enzyme reactions governed by the chemical step remains to be investigated further, but meta-analyses of kinetic databases show little correlation between $k_{cat}$ and $K_M$[26,28,43]. This is unlike many reactions in both homogenous and heterogeneous (non-bio-chemical) catalysis, which may be limited by an LFER even though the reaction is governed by a chemical step[44,45]. Kinetic

parameters for heterogeneous enzyme reactions are scarce. Thus, it is still an open question, whether scaling relations are as common in heterogeneous biocatalysis as they are in inorganic heterogeneous catalysis[46], but the current study shows that cellulases are severely restricted by an LFER.

**Consequences of the scaling relationship**. One aspect of the proposed scaling of $K_M$ and $k_{cat}$ is that the initial rate, $v_{ss}$ (Eq. 1), may be approximated by just one of the kinetic parameters. To illustrate this, we combined Eqs. 2–4 and solved for $k_{cat}$.

$$k_{cat} = AK_M^a \qquad (5)$$

In Eq. 5 $a = -\Phi$ and $A = \frac{k_{cat,ref}}{K_{M,ref}^{-\Phi}}$. Inserting Eq. 5 into the MM-equation (Eq. 1) expresses $v_{ss}$ as a function of $K_M$

$$v_{ss} = \frac{E_0 AK_M^a S_0}{S_0 + K_M} \qquad (6)$$

Equation 6 underscores, how ligand affinity is a double-edged sword. Hence, as demonstrated in the SI (supplementary note 3), Eq. 6 has a global maximum when $K_M$ attains the value

$$K_{M,opt} = S_0 \frac{a}{1-a} \qquad (7)$$

This implies that at a fixed load of substrate, $S_0$, a cellulase with low $K_M$ (i.e., $K_M < K_{M,opt}$) will become a better catalyst (increase $v_{ss}$) if it is engineered for weaker substrate binding. Conversely, weakly binding enzymes ($K_M > K_{M,opt}$) will gain from tighter binding. In the current case, $a = 0.74$ and insertion into Eq. (7) shows that $K_{M,opt} = 2.8 \ S_0$. In other words, the fastest initial rate on the current substrate (Avicel) will be observed for a cellulase that has a $K_M$ value that is around threefold higher than the Avicel load. To illustrate this, we plotted $v_{ss}$ as a function of $K_M$ for all of the investigated enzymes (excluding outliers identified in Fig. 2) at different substrate loads (Fig. 6). The results are in line with a previous observation[47] showing the so-called volcano plots, where cellulase activity tapers off on each side of the optimal affinity. Such volcano plots mirror the Sabatier principle, which states that the catalytic efficacy is optimal for a catalyst

with intermediate binding strength[48]. Higher/lower affinity leads to a situation where dissociation/association limits the overall rate. The optimal affinity, $K_{M,opt}$, depends on the substrate load and this is indicated by the black symbols in Fig. 6, which were calculated using Eq. 7. We emphasize that the appearance of an optimal $K_M$ is a direct consequence of the LFER, and that this type of analysis is well-established within (non-biochemical) heterogeneous catalysis[46,49].

As a final example of an application, we note that the LFER may be useful in computational selection and design of enzymes for technical use. Thus, a link between activity and affinity provides an important simplification as it converts the highly complex problem of in silico assessment of enzyme turnover frequency to the more tractable challenge of calculating binding energy. To illustrate this, we computationally assessed the strength of enzyme-substrate interactions for a subset of nine enzymes spread along the diagonal in Fig. 2. As shown in Fig. 3a, the computed binding energies scaled with the experimental values. These results suggest that the kinetic properties of novel, uncharacterized enzymes may be estimated by combining computed binding data with an experimental LFER based on a limited number of enzymes. Hence, efficient enzymes for a given set of experimental conditions could be identified through in silico screening.

In closing, the kinetic characterization of a wide group of fungal cellulases on their native, insoluble substrate revealed a LFER between substrate binding and activation barrier. We propose that this reflects basic physical restrictions of the hydrolytic reaction, which limits the evolutionary selection to a narrow lane around the scaling line, irrespectively of the enzymes' fold, modularity, or catalytic mechanism. The scatter around the proposed scaling line in Fig. 2 corresponds to a factor of about 2 in the value of $k_{cat}$. Hence, our results suggested that experimental $k_{cat}$ values for enzymes with approximately the same $K_M$ varied within this range. This variance encompassed a minor contribution from experimental errors, but it may also reflect kinetic diversity that results from differences in the mechanism and specificity of the tested enzymes. However, when we zoomed out and considered a broad range of $K_M$ values, this variance was modest, and the fitness landscape was dominated by a common scaling for all enzymes. Comparisons of wild types and variants revealed that small alterations in sequence (even point mutations) could lead to significant kinetic changes. In most cases, however, the changes involved a stringent movement on the scaling line rather than a shift away from the line, and this further demonstrated a strong coupling between affinity and turnover. We propose that this behavior is linked to the interfacial nature of the reaction. On one hand, strong ligand interactions are required to enable the transfer of a cellulose chain from the cellulose surface to the enzyme complex. On the other, a highly stable enzyme-substrate complex is inescapably associated with slow turnover (Figs. 5b–3). These relationships may help rationalize cellulolytic mechanisms and guide the selection of technical enzymes. It also appears that LFERs for interfacial enzyme reactions may establish a connection to (inorganic) heterogeneous catalysis, and hence pave the way for the use of practices and principles from this field within enzymology.

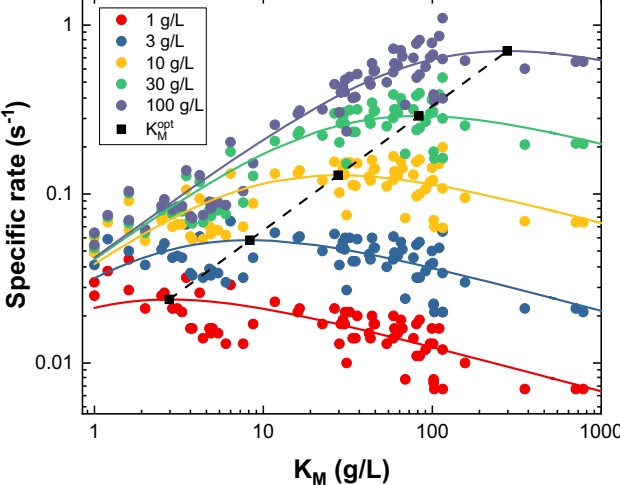

**Fig. 6 Volcano plots for five different substrate loads.** The specific rate at five different substrate loads is plotted as a function of $K_M$ for the investigated enzymes (excluding outliers identified in Fig. 2). Points represent experimental data and solid lines are the predicted volcano curves calculated using Eq. 6 and a = 0.74 (there are no free parameters in the determination of these solid curves). Black squares represent $K_{M,opt}$ values calculated using Eq. 7, and these points identify the maxima of the volcano plots at a given load of substrate.

## Methods

**Enzymes and kinetic measurements**. Experimental methods used in this work have been described elsewhere (see supplementary Table 4). Briefly, we expressed all enzymes heterologously in *Aspergillus oryzae* and purified as described elsewhere[50,51]. Engineered enzymes containing single or multiple amino acid substitutions, deletions or insertions was made using splicing overlap extension (SOE) PCR or by expression vector[50]. A full list of primers can be found in supplementary Table 5. For variants with added CBM, gBlocks™ Gene Fragments was ordered from Integrated DNA Technologies (IDT) overhang of 24 bp for SOE. SDS-PAGE gels (15-well NuPAGE 4–12% BisTris, GE Healthcare) revealed a single band for the purified enzymes, and their concentrations were determined by UV

absorbance at 280 nm using a theoretical extinction coefficient calculated based on amino acid sequence[52]. Michaelis–Menten (MM) curves were obtained as described previously[51] using 0.1 μM enzyme and microcrystalline cellulose (Avicel PH-101, Sigma-Aldrich, St. Louis, MO) load ranging from 1 to 100 g/L. MM curves were fitted to Eq. 1 in Origin pro v. 7. All experiments were done in triplicates at 25 °C using standard buffer 50 mM sodium acetate pH 5.0.

**Molecular dynamics simulation**. For simplicity, the modular structures of the different cellulases were split into two simulations. The CDs were simulated in complex with a cellononaose ligand and the CBMs (if present) were simulated bound to a cellulose crystal.

**Simulations of the catalytic domains**. If available, the structures were taken from the Protein Data Bank (TrCel7A: 4C4C[53], TrCel7B: 1EG1[54], TrCel6A: 1QK2[55], TrCel12A: 1H8V[56], TrCel45A: 3QR3[57], HiCel45A: 4ENG[58], Re7A: 3PL3). The ligand was inserted by alignment, if a related structure with a similar large disaccharide was available. Elsewise, docking with Autodock Vina was performed[59]. The ten clusters lowest in energy were inspected and the lowest energy configuration from the cluster with the closest distance between the catalytic residues and the glycosidic bond of interest was taken. The CHARMM36 force field was used to describe the system[60]. All simulations were run in GROMACS 2018.6[61]. Catalytic acids of all CDs were protonated. GROMACS was used to construct a cuboid box with edge lengths of 9.4 × 9.4 × 20 nm and the complexes were positioned at 4.7, 4.7, and 3.3 nm. The complexes were rotated so that the center of mass of the last and the fourth last sugar unit of the ligands were parallel to the z-axis. The systems were solvated with TIP3P water. To neutralize the net charges of the systems, random water molecules were exchanged with sodium ions. Minimization was conducted in a steepest-descent over 10'000 iterations. All subsequent simulations were performed at 300 K. NVT-simulations were performed for 100 ps while keeping the complex restraint. Thereafter, NPT-simulations with restraints on the solutes were performed for 100 ps. For all further simulations, only $C_\alpha$ further away than 1.5 nm from the ligand were restrained. A second round of NPT-simulations with the new restraints were performed for 100 ps. RMSD analysis of the protein backbone showed, that this time was sufficient to reach an equilibrated state. Thereafter, steered MD simulations were done over 800 ps with a pulling rate of 0.01 nm/ps and a force constant of 1000 kJ/mol/nm². The pull was performed on the first sugar unit of the cellononaose ligand in z direction. The resulting trajectories were used to prepare further simulations. Frames every travelled 0.5 Å by the ligand were extracted up to a final distance of 1 nm between the CD and the ligand. The extracted frames were used as starting configuration for Umbrella sampling simulation along the binding path. Each window was simulated for 620 ps, where the first 20 ps were disregarded as equilibration. It should be noted, that TrCel6A works from the opposite end compared to the other cellulases[21]. The set-up was adapted accordingly.

**Simulations of the carbohydrate-binding modules**. If available, the structures were taken from the Protein Data Bank (CBM1 of TrCel7A: 2CBH, CBM1 of TrCel7B: 4BMF). Otherwise, they were prepared through homology modelling by Modeller[62] (CBM1 of TrCel6A, CBM1 of TrCel5A, CBM1 of HiCel45A). A cellulose crystal of the type Iβ with a length of 5, a width of 6, and a depth of 3 unit cells was generated with the Cellulose Builder web server[63]. The CBMs were placed on the surface according to Beckham, et al[64]. A cubic box with a minimal distance of 1.0 nm was constructed. The crystal plane was oriented perpendicular to the z-axis. The simulations were performed in a similar fashion as the ones for the CD domains. However, the heavy atoms of the crystals were kept constrained after the energy minimization and the second NPT-simulation was increased to 1 ns to get the CBM settled on the crystal surface.

**Analysis**. Analysis of the trajectories was performed with GROMACS. The weighted histogram analysis method (WHAM) was applied to analyze the Umbrella sampling simulations along the binding path[65]. If density gaps occurred, additional windows at those distances were inserted iteratively until no gaps occurred. From the resulting PFM curves, the energy difference between the minimum and the maximum of those curves were taken. The errors were estimated with bootstrapping. Obtained $\Delta G_{B,MD}$ values from the CD and CBM part were added up to give values for the full enzyme. The energies were normalized by the values from the reference enzyme TrCel6A. This resulted in $\Delta\Delta G_{B,MD}$ values, which are more readily comparable to the experimental $\Delta\Delta G_{B,Exp}$. Linear regressions of the experimental binding energy and experimental activation energy against the computed binding energy were performed. The former resulted in a linear fit in the form of $y = 0.16x + 0.2$ and with a Pearson's coefficient $r^2 = 0.93$ and the later resulted in $y = 0.13x + 0.67$ with $r^2 = 0.81$. To counteract this known systematic overestimation issues of the method[66–68] and of the carbohydrate binding in general[69,70], a linear transformation on the initially obtained computed binding energies was performed using the parameters from the linear regressions. The final results for the prediction of the binding energies had a root-mean-squared error (RMSE) of 0.86 kJ/mol, the ones for the prediction of the activation energy had RMSE of 1.20 kJ/mol.

**Reporting summary**. Further information on research design is available in the Nature Research Reporting Summary linked to this article.

## Data availability
The authors declare that all the data supporting the findings of this study are available within the article (and Supplementary Information files), or available from the corresponding author on reasonable request. The following structures given by their PDB accession code was used in this study 2CBH, 4BMF, 4C4C, 1EG1, 1QK2, 1H8V, 3QR3, 4ENG and 3PL3. Source data are provided with this paper.

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

## Acknowledgements

This work was supported by Innovation Fund Denmark [Grant number: 5150-00020B], the Novo Nordisk Foundation [Grant number: NNF15OC0016606 and NNFSA170028392] and Independent Research Fund Denmark [Grant number: 8022-00165B].

## Author contributions

J.K., K.B. and P.W. planned and designed the study. G.A.M. characterized the enzymes, analyzed the data and interpreted the results along with J.K. T.H.S., A.M.C.M., N.S. and K.J. carried out cloning and expression of the enzymes. S.F.B., S.J.C., N.S.R., C.S.C., M.B.K., B.K., J.P.O., N.R.S. and K.B.R.M.K. purified the enzymes and performed initial biochemical experiments. C.S.C. performed the insoluble reducing ends measurements and control experiments. G.H.J.P. and K.S.S. did the MD simulations. J.K., G.A.M., K.S.S. and P.W. wrote the paper. All authors reviewed and approved the paper.

## Competing interests

T.H.S., J.P.O., K.B.R.M.K., K.J. A.M.C.M. and K.B. work for Novozymes a major manufacturer of industrial enzymes. All other authors declare no competing interests
