## [Peer Review File · Nature Communications]

REVIEWER COMMENTS

Reviewer #1 (Remarks to the Author):

The study from the Westh group is a monumental-in-scope and groundbreaking report on the application of linear free energy relationships to cellulases, which is commonplace in the heterogeneous catalysis community, but heretofore absent in the cellulase/interfacial biocatalysis community. Honestly I am surprised that this paper is not going to be published somewhere like Nature Catalysis, but it is absolutely appropriate for Nature Communications. This paper essentially should be published as fast as possible.

I have extremely minor points below. This is easily one of the most important papers (if not the most) in the cellulase field I've ever read or reviewed, and certainly the most exciting study I've seen since the original reports (including from Peter Westh, Priit Valjamae, and Kiyohiko Igarashi) that the rate-limiting step of GH7 CBH action is the off-rate. The development of a kinetic descriptor to screen enzymes for interfacial biocatalysis will have far-reaching implications in the cellulase field and well beyond.

As the authors may wish to point out in their Introduction with salient references, interfacial biocatalysis is getting quite hot these days in the plastics field as well. Might be worth pointing out!

There is a typo in the title; similarly throughout the authors drop the 't' in restraint and constraint.

I think the authors should be careful about calling GH7 CBHs "exo-acting" given that they have been shown to also conduct endo-initiation, e.g., see work from Jerry Ståhlberg and many others. Minor point, but worth noting.

Figure 1 – I tend to suggest including citations for the papers wherein PDB representations are provided. Ditto for Pymol (and all other programs that are considered 'methods').

The data in Figure 2 should be available in table format or machine readable format in the SI correlated to the specific enzymes.

The methods in the MD simulations for umbrella sampling absolute binding free energies (which I think is what the authors are doing? That is unclear to me) are fairly non-ideal for such large ligands. Did the authors "turn off" the ligand restraints/constraints at the end of the free energy cycle?

I'm sure that the authors know this, but literature from leaders in the field of free energy method development (e.g., Benoit Roux) should be examined; e.g., C Payne et al. in ref. 41 did similar work several years ago with H-REMD/FEP methods for GH7 CBHs – can the authors at least compare their results for the overlapping enzymes to that work? Perhaps though I am confused. I'm not sure how umbrella sampling provides a $\Delta\Delta G$ value, and not a ΔG value? That $\Delta\Delta G_{B_MD}$ value should be defined and a figure (main text or SI, doesn't matter) of the thermodynamic cycle from the MD simulations with the full constraints should be included – this is by far the most important comment of this review as it will help others to identify and reproduce what was done here.

Do the authors think that this kind of relationship would extend to LPMOs too?

Reviewer #2 (Remarks to the Author):

Kari et al have performed kinetic characterization of a large set of cellulases (83 enzymes) for release of soluble product from cellulose. They used the resulting parameters (k_{cat} , K_m for cellulose substrate) in linear free energy relationship (LFER) analysis. They observed that $\Delta\Delta G$ calculated from the k_{cat} were correlated linearly with $\Delta\Delta G$ calculated from the K_m . A number of (far-reaching) interpretations are made based on this type of LFER. The authors claim to have made important advance within the field (cellulose degradation) and to provide fundamental insight into principles of heterogeneous catalysis.

I do not agree with the authors' conclusions. While I find the study to have merit because of the extensive scope of enzymes used, the kinetic analysis is by far too simplistic. The interpretations are not clear and may even be flawed by failure to recognize, and properly disentangle, the fundamentally lumped nature of the kinetic parameters. I was unable to share the authors' excitement about their study contributing to a deepened understanding of catalysis at interfaces.

1) Study of catalytic processes at surfaces/interfaces is challenging. Cellulose hydrolysis by enzymes is one such process. It happens at the solid surface of the cellulose substrate and involves the additional feature that the solid is degraded. Mechanistically, the process requires multiple steps, several of which are physical and involve interaction at the solid surface (e.g. adsorption, molecular movement on the surface). Analysis of cellulase kinetics from the viewpoint of "interfacial enzymology" will have to consider the physical and biochemical events in a properly developed model/mechanism. The kinetic analysis of the current study is clearly oversimplified. It appears to be inapt to provide evidence of the sort required to support the claims made. The model used is formally of the Michaelis-Menten type. However, it is unclear how the model parameters relate to any of the mechanistic elements referred to in their extensive discussion. To be understood properly: this is not to say that the simple model cannot be used. But the interpretation must then respect the model's intrinsic limitations. Both the k_{cat} and the K_m are lumped composites of individual microscopic steps. The relevant dependencies are not resolved in the current study. Classical misinterpretation of K_m is that of a binding constant.

a) The LFERs show correlation between $\Delta\Delta G$ (k_{cat}) and $\Delta\Delta G$ (K_m). More simply stated, enzymes having a high k_{cat} (fast rate) tend to have a high K_m ("weak binding"). One can use various simple (one substrate-one product) enzyme mechanisms to show that this correlation is not very surprising, it may even be a necessity of the mechanism. The degree of correlation depends on the relative magnitude of rate constants for binding/dissociation and catalytic reaction. A strong linear correlation might be expected in a case where the actual binding is not much faster than catalysis. The point is this: if correct algebraic expressions for k_{cat} and K_m are derived based on a plausible/well supported enzyme mechanism, it will become immediately clear, deductively, what is the effect of change in k_{cat} (due to change of a certain rate constant) on change in K_m . It may turn out that the trend observed is just as expected (i.e. trivial) from the relevant mathematical relationships. The transformation into free energies does not alter anything about my fundamental concern. It will only serve to make an existing correlation appear more linear.

b) The correlation between $\Delta\Delta G$ (K_m) and computed binding energies is not clear. First, it is not clear if the K_m can be taken as a binding constant. Second, the binding energies (as far as I understand the methods used) describe the molecular interactions of the enzyme with a perfect surface of crystalline cellulose. What is the evidence that binding energies thus computed adequately reflect any of the experimentally accessible binding parameters?

c) If there is a proportional change in k_{cat} and K_m for the different enzymes (relative to the reference enzyme), the important ratio of k_{cat}/K_m cannot change much. I was wondering

what this can mean? I should be quite surprising if all enzymes had the same/similar efficiency or were lacking specificity compared one to another.

2) Considering the limitations pointed out, I find the discussion extremely speculative and for the most part insufficiently supported by the relevant evidence. For example, it is not clear what a fitness landscape for the cellulases should be (Figure 2 and 4). The discussion about uniform/differential binding or selective transition state or ground state binding also seems to lack a proper basis (Figure 5). It can be noted that the original papers on these effects in enzyme catalysis have been very rigorous and clear about the underlying enzymatic-kinetic mechanism. Further transformations (mathematical differentiation) of the simple hyperbola lead to correlations like that in Figure 6 which seem to indicate that enzymes of different K_m differ in catalytic rate depending on the substrate concentration used. I may misunderstand something, but the conclusion seems trivial to me.

In writing, the Abstract can represent various parts in the manuscript that are unclear because of overinterpretation. What are basic physical restrictions on the hydrolytic process? How has evolution "condensed" cellulases to operate along the line of the LFER? What is the link from heterogeneous inorganic catalysis to interfacial enzyme catalysis? I can follow the idea that prediction of the k_{cat} from a computed binding energy has appeal. However, to be relevant in practice, one would have to show that this applies to substrates different from the model cellulose used in their study.

Reviewer #3 (Remarks to the Author):

Kari et al submitted a very interesting paper that reports the discovery of a clear LFER among activation and binding free energy for a series of wt and engineered cellulases. The authors propose a new method that uses the LFER for optimizing cellulases and enzymes in general.

The paper is very clear and well written and the results are well supported by the data. The subject is of broad interest for the community even though its broad applicability is still not convincingly justified (see below). The article is very original and may inspire others to follow the author's path for enzyme optimization.

I recommend publication after the questions raised below are clarified.

General questions

g1. The most important aspect that is missing is a discussion on the transferability of the findings to other enzyme families. From a chemistry point of view, cellulases work all through the same chemical mechanism and cross similar transition states. The nuances are very minor. Do the authors expect the conclusions to other enzyme families? And across enzyme families? The former is crucial to measure the impact of the work. The latter is not so important for enzyme optimization but is interesting per se.

g2. If the kinetics is controlled by product release it is expectable that changes in binding free energy and activation free energy are correlated. In fact, the observed activation free energy may be a surrogate of the binding free energy of the product, which in turn is influenced by the binding free energy of the reactant.

A much more complex case emerges in enzyme families where the observed activation free energy comes from the chemical step.

The authors consider that the LFER might hold in that case? Why?

Minor points

m1. The discussion at lines 168-176 is a bit confusing. The authors divide fig. 2A into two regions - above and below the diagonal, with the first corresponding to low turnover and weak binding and the second to the opposite. However, the area above the diagonal includes regions of high turnover (SE) and regions of high affinity (NW). What it doesn't include is a region where both are favorable at the same time (i.e., the SW corner). The same comment (in the opposite direction) holds for the region below the diagonal.

m2. Concerning the physical origin of the correlation, the authors propose that it reflects an equilibrium between evolutionary selection and physical constraints.

This would mean that evolution drives the enzymes always in the way toward higher rates.

At ref. 26 (and in an upcoming paper including one thousand enzymes from all enzyme classes) we have seen that enzyme efficiency is narrowed to a sharp range, which suggests that enzymes may evolve not to work as fast as possible but instead at the right speed that maximizes overall fitness and the "concerted metabolism" of the organism. An excess of activity may break down homeostasis. Also, if we can evolve enzymes to make them to work faster this might mean that the physical constraints are not insurmountable. What the authors think about this?

Well, all of this are educated guesses, we don't really know the answers for these fascinating questions and I perfectly respect the opinion of the authors.

Pedro Alexandrino Fernandes.

Reviewer #1 (Remarks to the Author):

The study from the Westh group is a monumental-in-scope and groundbreaking report on the application of linear free energy relationships to cellulases, which is commonplace in the heterogeneous catalysis community, but heretofore absent in the cellulase/interfacial biocatalysis community. Honestly I am surprised that this paper is not going to be published somewhere like Nature Catalysis, but it is absolutely appropriate for Nature Communications. This paper essentially should be published as fast as possible.

I have extremely minor points below. This is easily one of the most important papers (if not the most) in the cellulase field I've ever read or reviewed, and certainly the most exciting study I've seen since the original reports (including from Peter Westh, Priit Valjamae, and Kiyohiko Igarashi) that the rate-limiting step of GH7 CBH action is the off-rate. The development of a kinetic descriptor to screen enzymes for interfacial biocatalysis will have far-reaching implications in the cellulase field and well beyond.

We thank the reviewer for the positive feedback and constructive comments. In the following, we have made a point-by-point response to the comments made by the reviewer. To increase readability of our response, comments are shown in red, while our reply is shown without any change in font or color. We highlighted, added or changed text in the main manuscript or SI in green and *italic*. Original text from the manuscript is shown in *italic*.

As the authors may wish to point out in their Introduction with salient references, interfacial biocatalysis is getting quite hot these days in the plastics field as well. Might be worth pointing out!

We thank the reviewer for the comment. We have now added the following small change in the introduction (page 2):

...but more commonly, the activity of soluble enzymes on insoluble substrates such as polysaccharides, lipids, precipitated proteins (5) or more recently plastic (6).

There is a typo in the title; similarly throughout the authors drop the 't' in restraint and constraint.

We have corrected the mentions typo's in the main text.

I think the authors should be careful about calling GH7 CBHs "exo-acting" given that they have been shown to also conduct endo-initiation, e.g., see work from Jerry Ståhlberg and many others. Minor point, but worth noting.

We agree with the reviewer and have removed the word "exo-acting" from the text.

Figure 1 – I tend to suggest including citations for the papers wherein PDB representations are provided. Ditto for Pymol (and all other programs that are considered 'methods').

Good point. We have now added references to all PDB structures used in the main text and to all programs used in this study.

The data in Figure 2 should be available in table format or machine readable format in the SI correlated to the specific enzymes.

We have uploaded an Excel file with this information in addition to the SI.

The methods in the MD simulations for umbrella sampling absolute binding free energies (which I think is what the authors are doing? That is unclear to me) are fairly non-ideal for such large ligands. Did the authors “turn off” the ligand restraints/constraints at the end of the free energy cycle?

The condensed description of the method for the MD simulation have now been extended in the main text and elaborated in the SI (see answer below) to clarify how the binding energies were computed. Since binding energies were computed using steered MD with umbrella sampling we did not compute the energies from a thermodynamic cycle. The method was earlier employed for computing the binding energies of small peptides in amyloid fibrils (1), therefore we do not think the size of the ligands is per se an issue (of course big, flexible molecules are never ideal). During the umbrella sampling simulations, restraints to keep the ligand in a relative position along the binding path were used (see illustration details below).

I’m sure that the authors know this, but literature from leaders in the field of free energy method development (e.g., Benoit Roux) should be examined; e.g., C Payne et al. in ref. 41 did similar work several years ago with H-REMD/FEP methods for GH7 CBHs – can the authors at least compare their results for the overlapping enzymes to that work? Perhaps though I am confused. I’m not sure how umbrella sampling provides a $\Delta\Delta G$ value, and not a ΔG value? That $\Delta\Delta G_{B_MD}$ value should be defined and a figure (main text or SI, doesn’t matter) of the thermodynamic cycle from the MD simulations with the full constraints should be included – this is by far the most important comment of this review as it will help others to identify and reproduce what was done here.

The steered MD with umbrella sampling were chosen over other methods such as free energy perturbation, since the binding path is fairly well determined for cellulases as opposed to the alchemical path of thermodynamic cycle of the system, which in the current set-up would involve perturbing a cellulose crystal or the CBM.

In this study we only report changes in the binding energies ($\Delta\Delta G_{MD}$) although the output from the simulations were absolute binding energies. We normalized the binding energies with that of TrCel6A for two reasons:

- 1) Absolute computed binding energies of cellulases tend to overestimate the binding strength compared to the experimental values.
- 2) Changes in binding energies from the MD simulations ($\Delta\Delta G_{MD}$) scaled with binding strength derived from experimental values (See fig. 3, main text)

Payne et al. (2) computed the ligand binding free energy of the catalytic domain of 5 GH7 cellulases. For TrCel7A these authors found a binding energy of 116.4 ± 8.8 kJ/mol (-27.8 ± 2.1 kcal/mol) for TrCel7A in complex with cellononaose. In comparison, we found an absolute binding energy for the TrCel7A-cellononaose complex of -73.9 ± 2.5 kJ/mol. This binding strength is somewhat weaker than Payne et al. (2), but stronger than what have been found experimentally (3).

To clarify the computational approach used in this study, we have added the following figure to the SI and made changes in the main text (see below).

Figure S2. Illustration of the workflow used to compute the binding energies in this study. The binding energies were derived from steered MD simulation with umbrella sampling along the binding path. This was done in parallel for the catalytic domain and the carbohydrate-binding module (if present) and the simulations were analyzed using weighted histogram analysis method. In order to compare with the experimental result the obtained absolute binding energies were normalized with TrCel6A (ΔG_{ref}) to obtain the change in binding free energies ($\Delta\Delta G_{MD}$) which could be compared to the experimental derived energies $\Delta\Delta G_{exp}$.

We have added the following text in the result section (main text, page 6):

...with umbrella sampling along the ~~reaction~~ binding path (See further details in figure S8, S1).

And in the method section (page 13) we added the following.

...The errors were estimated with bootstrapping. Obtained ΔG_B values from the CD and CBM part were added up to give values for the full enzyme. The energies were normalized by the values from the reference enzyme TrCel6A. This resulted in $\Delta\Delta G_{B,MD}$ values, which are more readily comparable to the experimental $\Delta\Delta G_{B,Exp}$. A linear regressions of...

And in the same section (method section, page 13) we added.

....To counteract this known systematic overestimation issues of the method (64-66) and of the carbohydrate binding in general (67,68), a linear transformation on ...

Do the authors think that this kind of relationship would extend to LPMOs too?

It is an interesting question. We have not tested any LPMO in the current study since the activity of these enzymes are difficult to measure experimentally. Their activity mostly give raise to new insoluble ends, which complicate the analysis. However, synergy between different cellulases including LPMOs is probably a way to escape the observed LFER. We are currently investigating these effect but further studies will be needed to answer the question properly.

Reviewer #2 (Remarks to the Author):

Kari et al have performed kinetic characterization of a large set of cellulases (83 enzymes) for release of soluble product from cellulose. They used the resulting parameters (k_{cat} , K_M for cellulose substrate) in linear free energy relationship (LFER) analysis. They observed that $\Delta\Delta G$ calculated from the k_{cat} were correlated linearly with $\Delta\Delta G$ calculated from the K_M . A number of (far-reaching) interpretations are made based on this type of LFER. The authors claim to have made important advance within the field (cellulose degradation) and to provide fundamental insight into principles of heterogeneous catalysis. I do not agree with the authors' conclusions. While I find the study to have merit because of the extensive scope of enzymes used, the kinetic analysis is by far too simplistic. The interpretations are not clear and may even be flawed by failure to recognize, and properly disentangle, the fundamentally lumped nature of the kinetic parameters. I was unable to share the authors' excitement about their study contributing to a deepened understanding of catalysis at interfaces.

We appreciate the reviewer's criticism and input regarding the kinetic analysis. Indeed, these comments have served as inspiration for the introduction and analysis of a novel kinetic model, as well as the inclusion of new experimental investigations in the revised manuscript. These new contributions are described over the following pages. On a general level, we claim that the combination of extensive kinetic data on one hand and a simple model on the other makes up the most promising way forward for cellulase biochemistry at the current stage. This is partly because our understanding of structure/function relationships for many cellulases is too limited to support formulation of sophisticated reaction schemes. Moreover, current assay methodologies for interfacial enzyme reactions are limited and the experimental requirements to support the analysis of complex models (*i.e.* to resolve many rate constants) can hardly be met. Finally, the general idea of describing a complex catalysis through simplified (lumped) parameters has previously proven very effective within non-biochemical heterogeneous catalysis. In this latter field, complex reactions with multiple elementary steps have been shown to be adequately described by a few apparent parameters, which are good descriptors of the overall catalyzed reactions (4). The success of simplifying these inorganic reactions relies on numerous LFERs that link the real, elementary rate constants and the overall motivation for the current work is to see if similar shortcuts could be applicable within interfacial enzymology.

Our specific responses to the reviewer's queries as well as the description of a new kinetic model and additional experiments turned out to be quite long. Hence, to facilitate reading, we have organized our reply in three parts

1. **The model:** Critique regarding the application of the MM-model for cellulases
2. **The parameters:** Critique regarding the interpretation of the kinetic parameters (K_M and k_{cat})
3. **The LFER:** Critique regarding the interrelationship between the kinetic parameters (K_M and k_{cat})

A detailed point-by-point answer can be found over the following pages. To increase readability of our response, comments are shown in red, while our reply is shown without any change in font or color. We highlighted, added or changed text in the main manuscript or SI in green and *italic*. Original text from the manuscript is shown in *italic*.

First, however, we summarize the contents and main conclusions of the three parts.

Summary

1) *The model.* We tested the idea of the reviewer that the model used in this study was too simple for the type of system studied here. Specifically, we derived a novel steady-state rate-equation for a detailed model of the best studied cellulase (TrCel7A) to elucidate the consequence of the simple MM-approach. We showed that the steady-state equation of the complex model can be expressed as a MM-equation and that the apparent MM-parameters for the complex model only gave rise to a scaling relationship when the rate-constant for the unbinding was changes (similar to what was suggested in the main text). The kinetic modelling show that the simple MM-scheme may be regarded as a composite reaction scheme for the system studied here. This support the use of the MM-model for comparative analysis of a large set of cellulases where most of the enzyme have no or little information about their molecular steps and underlying rate-constants.

The new model is introduced and analyzed in the revised SI, and we made reference to this on page 4 of the main text.

...and shown that Eq. 1 is valid and applicable even though the substrate is solid and specified by its mass load (S0) in units of g/L (32,33). The relationship between parameters from a more realistic model and those from the simple MM framework is further discussed in the SI. The derived rates were based...

2) *The parameters.* Since the MM-model is a simplification of the system, the MM-parameters will be composite parameters. In the main text we used apparent K_M as a descriptor for the enzyme-substrate affinity and k_{cat} as a descriptor for the dissociation step (rate-constant for the unbinding). This method was criticized by the reviewer and to test the assumption beyond the arguments in the main text, we conducted extensive additional experiments elucidating the binding kinetics of a selected group of cellulases from fig. 2 (main text). The result showed that activation free energy of dissociation scaled with the free energy of binding. Since this new data presented true rate- and dissociation constants (k_{on}, k_{off}, K_D) the result strongly support the kinetic analysis used in the main text and the energies derived from the apparent kinetic parameters.

To support the kinetic analysis and interpretation of the MM-parameters we added a section in the SI with the results from the new binding kinetic experiments. We made reference to this material on page 4 of the main text:

...We used k_{cat} and K_M , to estimate changes in respectively transition-state free energy ($\Delta\Delta G^\ddagger$) and standard free energy of ligand binding ($\Delta\Delta G_B^0$) (c.f. Fig. 5) following well-established principles (34,35). The value of K_M as descriptor of the enzyme-substrate affinity is further substantiated in the SI (see figure S6, SI). Specifically, we used the equations...

3) *The LFER.* The reviewer was concerned that the reported LFER was trivial and further questioned whether the results were applicable to other cellulosic substrate than the one used in this study. To test this idea we searched the literature and found that scaling between k_{cat} and K_M is not typical for hydrolases. Further, we conducted additional experiments (MM-curves) for a representative group of cellulases from fig.2 (main text) on another cellulosic substrate (amorphous cellulose) and found similar scaling between the binding energies and activation energies. This result show that the LFER is non-trivial and that it might hold for many types of cellulose substrates.

Detailed answers:

1) *The model*

1) Study of catalytic processes at surfaces/interfaces is challenging. Cellulose hydrolysis by enzymes is one such process. It happens at the solid surface of the cellulose substrate and involves the additional feature that the solid is degraded. Mechanistically, the process requires multiple steps, several of which are physical and involve interaction at the solid surface (e.g. adsorption, molecular movement on the surface). Analysis of cellulase kinetics from the viewpoint of “interfacial enzymology” will have to consider the physical and biochemical events in a properly developed model/mechanism. The kinetic analysis of the current study is clearly oversimplified. It appears to be inapt to provide evidence of the sort required to support the claims made. The model used is formally of the Michaelis-Menten type. However, it is unclear how the model parameters relate to any of the mechanistic elements referred to in their extensive discussion. To be understood properly: this is not to say that the simple model cannot be used. But the interpretation must then respect the model’s intrinsic limitations. Both the k_{cat} and the K_m are lumped composites of individual microscopic steps. The relevant dependencies are not resolved in the current study. Classical misinterpretation of K_m is that of a binding constant.

The main critique from the reviewer is that the Michaelis-Menten equation used to analyse the steady-state kinetics is too simple for the type of system studied here. As pointed out by the reviewer, cellulase kinetics is complicated, since it involves multiple elementary steps at the solid-liquid interface of an insoluble substrate. The best studied cellulase is the processive cellobiohydrolase from the fungus *Trichoderma Reesei* (TrCel7A) and much is known about its elementary steps in the overall hydrolysis of cellulose (5). We will use this enzyme as an example of how a detailed mechanism may be condensed into a Michaelis-Menten type of model and from this induce the generality of the MM-model for the other cellulases.

The molecular steps for TrCel7A have been reviewed in detail by Payne et al. (6) and include (see also figure S1):

1. Adsorption of the enzyme to the cellulose surface
2. Threading of an accessible free cellulose chain end into the active site tunnel
3. Processive catalysis, which includes 3.1) hydrolysis of the glycosidic bond, 3.2) product expulsion from the product site and 3.3) sliding to form the next Michaelis-complex.
4. Dissociation from the surface

Figure S1. Schematic illustration of the different molecular steps involved in the processive hydrolysis of cellulose by cellobiohydrolases.

Step 1 and 2 in figure S1 have recently been investigated for TrCel7A (7,8). In these studies, time-resolved measurements of the threading and adsorption were used to elucidate the rate constants for the

formation of a productive enzyme-substrate complex. In both cases, the threading/dethreading (step 2 in figure S1) was found to be slow compared to the fast dynamic of the adsorption/desorption step (step 1 in figure S1). This implies that the adsorbed but unthreaded enzyme rapidly inter-convert compared to other (slower) steps in the reaction mechanism and hence may be assumed to be in (rapid) equilibrium with respect to the threaded enzyme complex at all time. The next step (step 3 in figure S1) is what has been termed the *processive cycle* (9). This step is actually not one but several elementary steps, which a processive cellulase cycle through each time it moves productively (produces product) in the forward reaction path. The speed at which TrCel7A moves forward in its processive cycle has been measured to 7 nm/s by high-speed AFM (10). Since the product of TrCel7A is cellobiose, with a length of 1 nm, this translates into an overall rate constant for the processive cycle of 7 s^{-1} , which also matches values found from pre-steady-state measurement (11,12). In a recent study by Knott et al. (13) it was shown that the slowest step in the processive cycle of TrCel7A was the actual bond cleavage. By transition state path sampling, the authors found that the first step (glycosylation) in the two-step retaining catalytic mechanism, was the slowest step with a theoretical rate constant of 11 s^{-1} . This is remarkably close to the 7 s^{-1} found by Igarashi et al. (10) and supports the hypothesis that the actual bond cleavage is the slowest step in the processive cycle (step 3 in figure S1). Since the processive cycle consist of a sequence of first-order elementary steps, the slowest step (glycosylation) will dominate at steady-state (14) and we will call this step *catalysis*. Due to the processive nature of the enzyme, the *catalysis* “step” is repeated n times, where n is the processivity number of the enzyme. It has been shown that for TrCel7A, n is limited by the substrate (15) since the experimentally determined processivity is much smaller than the inherent processivity of the enzyme. This phenomenon may also be seen from pre-steady-state measurements, where the build-up of unproductive enzymes give rise to so-called burst kinetics (11,12,16-18). Real-time imaging of TrCel7A have shown that the build-up of unproductive enzyme is due to irregularities on the substrate that hinder further processive movement (10).

Based on the information above we can write a microkinetic scheme for the most important elementary steps in the overall hydrolysis of cellulose by TrCel7A (see scheme S1).

In table S1 we have summarized previously determined rate-constant for all steps in scheme S1.

In the following, we will try to illustrate how the more realistic, but complicated model would affect the apparent parameters and how scaling between apparent K_M and k_{cat} is restricted to the dissociation/dethreading event.

Table S1. Experimentally determined rate-constants and processivity number for TrCel7A. The rate-constants refer to the reaction steps defined in scheme S1.

Reaction steps	Rate constant	reference
Adsorption	$k_1 \sim 0.1 \text{ s}^{-1} \mu\text{M}^{-1}$	(7)
Desorption	$k_{-1} \sim 0.1 \text{ s}^{-1}$	(7)
Threading	$k_2 \sim 0.05 \text{ s}^{-1}$	(7,8)
Dissociation/Dethreading	$k_{-2} \sim 0.01 \text{ s}^{-1}$	(7,11)
Processive catalysis	$k_3 \sim 5 \text{ s}^{-1}$	(11,19)
Parameter		

We derived a steady-state rate equation for the reaction scheme S1 using the two-Step computer-assisted method developed by Fromm and Fromm (21). In the derivation, we assumed rapid equilibrium with the free and absorbed enzyme (step 1) which (as explained above) has been found to be a valid approximation. We also assumed that the substrate was in excess over the enzyme ($S_0 \gg E_0$) and that the total enzyme concentration was constant (mass conservation);

$$E_0 = E + ES_{adsorb} + ES_{thread} + ES_{np}$$

The differential equation was setup and solved in Matlab using the two-Step computer-assisted method and the result was the steady-state rate equation given in eq. S1

$$v = \frac{E_0 S_0 \frac{nk_{-2}k_3k_2}{(k_3 + k_{-2})(k_2 + k_{-2})}}{S_0 + \frac{k_{-1}k_{-2}(k_3 + k_{-2}) + k_3k_2k_{-2}}{k_1(k_3 + k_{-2})(k_2 + k_{-2})}} \quad (\text{Eq. S1})$$

Equation eq. S1 can be simplified to a Michealis-Menten type of equation (eq. S2)

$$v = \frac{E_0 S_0^{app} k_{cat}}{S_0 + {}^{app}K_M} \quad (\text{Eq. S2})$$

Where

$${}^{app}k_{cat} = \frac{nk_{-2}k_3k_2}{(k_3 + k_{-2})(k_2 + k_{-2})} \quad (\text{Eq. S3})$$

$${}^{app}K_M = \frac{k_{-1}k_{-2}(k_3 + k_{-2}) + k_3k_2k_{-2}}{k_1(k_3 + k_{-2})(k_2 + k_{-2})} \quad (\text{Eq. S4})$$

Since the dethreading step is known to be much slower than both the catalytic step and the threading step (see table S1), we may simplify the apparent MM-parameters

$${}^{app}k_{cat} \sim nk_{-2} \quad \text{for } k_{-2} \ll k_2 \wedge k_3 \quad (\text{Eq. S5})$$

$${}^{app}K_M \sim \frac{k_{-2}(k_{-1} + k_2)}{k_2k_1} \quad \text{for } k_{-2} \ll k_2 \wedge k_3 \quad (\text{Eq. S6})$$

As seen from eq. S5 and S6, the only rate-constant that is present in both the apparent k_{cat} and K_M is the rate-constant for the dethreading. Hence, direct scaling between the two apparent MM-parameters will be restricted to changes in the rate of dethreading. To test this prediction, we also simulated eq. S1 (with no additional assumption), where each parameter in the equation was changed to a value 10-times higher or 10-times lower than the initial value. The initial value was the most likely parameter extracted from literature (see table S1). A total of 10 values were scanned for each parameter and the step size between each value was logarithmic. The result of this analysis is shown in figure S2.

Figure S2. Sensitivity analysis of Eq. S1. Equation S1 was simulated as a function of substrate concentration ($0 \rightarrow 10 \mu\text{M}$) using the parameters given in table S1 as initial input. The enzyme concentration was constant ($0.1 \mu\text{M}$) in all simulations. Each of the 6 subplot shows how the rate equation changes when all parameters are kept constant, except the one given by the title of the subplot. Each parameter was scanned individually with 10 different values (log-scaled step size) spanning from 10 times smaller to 10 times larger than the value reported in table S1.

Since the steady-state solution (eq. S1) can be expressed as a simple MM-equation (eq. S2) we fitted the MM-equation to all of the curves shown in figure S2 to obtain apparent MM-parameters for all the $6 \times 10 = 60$ saturation curves. The apparent k_{cat} (V_{max}/E_0) and K_M is shown in figure S3.

Figure S3. All curves in figure S2 were fitted to a Michaelis-Menten equation (Eq. S2) and the derived parameters (K_M and V_{max}/E_0) obtained from the non-linear regression analysis are plotted as bar plot. For each of the 6 parameters there are 10 bars, representing the magnitude of K_M (lower panel) and V_{max}/E_0 (upper panel) as the parameter was changed.

Figure S4. Correlation plot between V_{max} and K_M . The values shown in figure S3 were plotted for each of the 6 parameters. To illustrate the connection to the LFER reported in the main manuscript, the parameters were normalized and log scaled. We normalized all parameters with the K_M and V_{max} values, obtained from a simulation using the parameters from table S1.

As seen from figure S3, only the processivity (n) and the dethreading had significant impact on the apparent k_{cat} . This result is in accordance with another model for processive cellulase (22) and experimental data (23), which show that TrCel7A and possibly other cellulases have a low turnover

number due to a low dissociation rate. From figure S3 it may also be seen that the apparent K_M is more complicated to interpret. However, only the change in k_{-2} (rate constant for the dethreading) gave rise to a linear scaling between the apparent K_M and the apparent k_{cat} . This point is illustrated in figure S4 where the normalized parameters are plotted on a logarithmic scale. Figure S4 serve to show that although the apparent K_M may be influenced by several of the rate-constants in scheme S1, only change in the dethreading give scaling as the observed scaling described in the main text.

To emphasize this point we have added a discussion of the simple model in the SI. In the main text (page 4, result section) we added the following.

...and shown that Eq. 1 is valid and applicable even though the substrate is solid and specified by its mass load (S_0) in units of g/L (32,33). The relationship between parameters from a more realistic model and those from the simple MM framework is further discussed in the SI. The derived rates were based...

2) The parameters

a) The LFERs show correlation between $\ln(k_{cat})$ and $\ln(K_M)$. More simply stated, enzymes having a high k_{cat} (fast rate) tend to have a high K_M ("weak binding"). One can use various simple (one substrate-one product) enzyme mechanisms to show that this correlation is not very surprising, it may even be a necessity of the mechanism. The degree of correlation depends on the relative magnitude of rate constants for binding/dissociation and catalytic reaction. A strong linear correlation might be expected in a case where the actual binding is not much faster than catalysis. The point is this: if correct algebraic expressions for k_{cat} and K_M are derived based on a plausible/well supported enzyme mechanism, it will become immediately clear, deductively, what is the effect of change in k_{cat} (due to change of a certain rate constant) on change in K_M . It may turn out that the trend observed is just as expected (i.e. trivial) from the relevant mathematical relationships. The transformation into free energies does not alter anything about my fundamental concern. It will only serve to make an existing correlation appear more linear.

The reviewer states that *"... enzymes having a high k_{cat} (fast rate) tend to have a high K_M ("weak binding"). One can use various simple (one substrate-one product) enzyme mechanisms to show that this correlation is not very surprising, it may even be a necessity of the mechanism."*

We do not agree with the reviewer that this relationship is trivial. There does not seem to be any a priori reason to expect so and certainly no reason to make this conclusion before it is experimentally substantiated. If, for example, we look at published meta-analyses of databases for kinetic parameters (for conventional bulk enzyme reactions) no clear picture of K_M/k_{cat} relations occur. Bar-Even et al. (24) analyzed several thousand enzymes and concluded "we find only a minor global correlation between k_{cat} and K_M ($R^2 = 0.09$)". Sousa et al. (25) reached the same conclusion in a recent study *"Interestingly, $\Delta G_{cat} \neq$ do not seem to correlate with ΔG_{bind} for hydrolases acting on C-N bonds ($r^2 \sim 0$) and only weakly for the other hydrolase subclasses ($0.1 < r^2 < 0.3$)"*.

The results from the simulation in the previous section ("The model") show that scaling between the apparent K_M and V_{max} solely exist for the rate equation eq. S1 when the dethreading is changed (figure

S4). We agree with the reviewer that the apparent MM parameters should be interpreted with caution. For this reason, we made additional experiments to justify our analysis of the parameters.

Measurement of the complexation kinetics

The complicated reaction scheme S1 of TrCel7A, can be significantly simplified if the concentration of threaded complex (ES_{thread}) can be measured experimentally. In such case, the reaction can be represented by a simple binding scheme (scheme S2) where the rate-constants k_{on} and k_{off} govern the threading and dethreading step respectively.

We have recently developed such method for cellulases (8) based on intrinsic fluorescence and we selected three WT cellulases for this new analysis. The three WT cellulases were TrCel6A, TrCel7B and HiCel45A of the main manuscript. Together with already published data for TrCel7A and four variants of TrCel7A (8) these enzymes cover a broad spectrum of affinity and structural diversity of cellulases. In the additional experiments conducted here we made real-time measurements of the change in fluorescence upon complexation with different loads of amorphous cellulose (RAC), as described in detail elsewhere (8). The experiments could unfortunately not be done using Avicel as substrate since the large particle size gave rise to excessive light scattering.

The raw-data (see figure S5A) was fitted to a single exponential function (Eq. S7) with a constant term representing the steady-state fluorescence signal (F_{ss}).

$$F(t) = F_{ss} + A e^{-t/b} \quad (\text{Eq. S7})$$

The steady-state signal (F_{ss}) was fitted to a single-site binding isotherm (Eq. S8) to determine the dissociation constant (K_D) and the maximal fluorescence signal (F_{max}), which is obtained when all enzymes are threaded.

$$F_{ss}(S_0) = \frac{F_{max} S_0}{S_0 + K_D}, \quad K_D = \frac{k_{off}}{k_{on}} \quad (\text{Eq. S8})$$

The threading rate (v_{on}) can now be estimated from the initial slope of the curves in figure S5A. We estimated this rate from the derivative of Eq. S7 solved for $t=0$ ($F'(t=0) = -\frac{A}{b} e^{-0/b} = -\frac{A}{b}$).

Since $ES \rightarrow E_0$ for $t \rightarrow 0$ we can write the on-rate as shown in equation S9

$$v_{on} = E_0 S_0 k_{on} = E_0 \frac{-A}{F_{max} b} \quad (\text{Eq. S9})$$

Where E_0 and S_0 are the initial concentration of enzyme and substrate and A , b and F_{max} are fitting parameters from respectively equation S7 and S8.

Equation S7 was fitted to the raw data shown in figure S5 to obtain the parameters A , b and F_{ss} . The latter parameter was plotted against the substrate load and fitted to equation S8 (figure S5B). Using

equation S9 and the derived parameters A , b and F_{\max} we could plot the on rate (v_{on}) as a function of substrate load (S_0) for the three cellulases (figure S5C).

The bi-molecular rate-constant k_{on} was derived from the slope of the curves in figure S5C and k_{off} was derived using from the K_D values ($k_{off} = K_D * k_{on}$). The rate-constants for the three enzymes, as well as recently published data are given in table S2.

Figure S5. A) Real time fluorescence data for the complexation of RAC and respectively TrCel7B, TrCel6A and HiCel45A (250 nM enzyme). The ordinates show fluorescence emission in arbitrary units, and is related to the fraction of enzyme molecules in the threaded (ES_{thread}) complex. Black lines represent best fit to the exponential function eq. S7. The final RAC load in each sample appear from the legends. B) Steady-state fluorescence emission at different RAC loads. Solid lines represent best fit to eq. S8. C) Rate of threading (v_{on}) for different RAC load. The threading rate was calculated using eq. S9. The parameters in equation S9 were obtained from the non-linear regression analysis shown in panel A and figure B.

Table S2. Parameters derived from the data shown in fig. S5 and published data by Røjel et al. (8).

Enzyme	Mode of action	CBM	k_{on} (L/g s^{-1})	$k_{off} \cdot 10^{-3}$ (s^{-1})	K_D (mg/L)	Reference
TrCel7B	EG	+	0.80 ± 0.09	79.3 ± 14.4	99.2 ± 14.1	
HiCel45A	EG	+	0.47 ± 0.05	32.8 ± 7.5	70.0 ± 14.2	Fig. S5
TrCel6A	CBH	+	0.40 ± 0.02	43.4 ± 16.4	105.6 ± 39.6	

TrCel7A	CBH	+	0.59 ± 0.01	4.8 ± 1.3	8.2 ± 2.2	
TrCel7A _{CD}	CBH	-	0.33 ± 0.02	5.9 ± 2.8	18.2 ± 8.3	
TrCel7A _{W38A}	CBH	+	0.41 ± 0.06	16.6 ± 8.1	40 ± 19.2	(8)
TrCel7A _{W40A}	CBH	+	0.39 ± 0.01	9.4 ± 3.0	23.8 ± 7.4	
TrCel7A _{W376A}	CBH	+	0.43 ± 0.06	2.2 ± 0.7	5.2 ± 1.5	

To further corroborate the analysis used in the main article, we calculated the change in binding free energy and activation free energy for the complexation as in the main text (eq. 2 and eq. 3, manuscript), where K_M and k_{cat} are substituted with respectively the dissociation constant for the complexation (K_D) and the rate-constant for either the threading (k_{on}) or dethreading (k_{off}).

$$\Delta\Delta G_B = RT \ln\left(\frac{K_D}{K_{D,ref}}\right) \text{ (eq. S10)}$$

$$\Delta\Delta G^\ddagger = -RT \ln\left(\frac{k}{k_{ref}}\right) \text{ (eq. S11)}$$

Similar to the main manuscript, we have in eq. S10 and S11 normalized all energies with respect to TrCel6A, a GH6 cellobiohydrolase from *Trichoderma reesei*. The scaling between the activation energy and binding energy is shown in fig. S6 for all enzymes in table S2. As seen from the figure, the change in the activation energy for the dethreading step ($\Delta\Delta G_{off}^\ddagger$) scales with the change in binding free energy with a slope of -1.05 ± 0.10 . This observation is similar to the finding in the main manuscript where the binding energies and activation energies derived from the Michaelis-Menten parameters scaled with a slope of -0.74 ± 0.02 . This independent set of experiments, which provides direct measurements of k_{on} and K_D , support our interpretation of the apparent MM-parameters and justify the use of eq. 2 and 3 (main text) to derive binding energies and activation energies.

Figure S6. Correlation plot of change in the binding free energy ($\Delta\Delta G^{\circ}_b$, Eq. S10) and activation free energy ($\Delta\Delta G^{\ddagger}$, Eq. S11) for all enzymes shown in table S2. The blue and red symbols respectively show how the activation free energy for the threading ($\Delta\Delta G^{\ddagger}_{on}$) and dethreading ($\Delta\Delta G^{\ddagger}_{off}$) changes with binding ($\Delta\Delta G^{\circ}_b$). Solid lines derive from the linear regression to the experimental data.

To support the kinetic analysis and interpretation of the MM-parameters we added a section in the SI with the results from the binding kinetic experiments. In the main text (page 4, result section), we added the following

...We used k_{cat} and K_M to estimate changes in respectively transition-state free energy ($\Delta\Delta G^{\ddagger}$) and standard free energy of ligand binding ($\Delta\Delta G^{\circ}_b$) (c.f. Fig. 5) following well-established principles (34,35). The value of K_M as descriptor of the enzyme-substrate affinity is further substantiated in the SI (see figure S6, SI). Specifically, we used the equations...

3) The LFER

b) The correlation between $\Delta\Delta G$ (K_M) and computed binding energies is not clear. First, it is not clear if the K_M can be taken as a binding constant. Second, the binding energies (as far as I understand the methods used) describe the molecular interactions of the enzyme with a perfect surface of crystalline cellulose. What is the evidence that binding energies thus computed adequately reflect any of the experimentally accessible binding parameters?

The threading kinetics for a representative subset of the investigated enzymes (see point 2) show that the binding energy ($\Delta\Delta G^{\circ}_b(K_D)$) and activation energy of the dethreading ($\Delta\Delta G^{\ddagger}(k_{off})$) scale similarly as the apparent MM-parameters. Unfortunately, the threading experiments could not be done using Avicel as substrate since the large particle size gave rise to excessive light scattering. For this reason, we also made Michaelis-Menten curves for the enzymes given in table S2 on RAC to test if the scaling between K_M and k_{cat} was also found on RAC. The results (derived apparent MM-parameters) of these experiments are plotted in figure S7 together with data for the same enzymes on Avicel (Avicel data was taken from the main article). The two EG's (TrCel7B and HiCel45A) showed significantly higher activity than the other enzymes on RAC (see open symbols in fig. S7), but otherwise similar scaling between K_M and k_{cat} was observed for RAC and Avicel. The origin of the high activity of the EGs on RAC needs further experiments to be elucidated but it is interesting to notice that the maximal steady-state rate (V_{max}/E_0) of these enzymes on RAC is close to the reported k_3 (see scheme S1) for GH7 enzymes. As discussed in the main text, the scaling between K_M and V_{max}/E_0 probably occurs because the rate-limiting step is not the catalytic event, but dissociation. It follows that the scaling will eventually break down for enzymes with fast off-rates or enzymes that can escape the enzyme-substrate complex by hydrolysis. In either case the observed maximal steady-state rate (V_{max}/E_0) would approach a new rate-limiting step, which most likely would be the actual bond breaking.

Another interesting observation from figure S7 is that linear regression to the experimental derived parameters (excluding the two EGs on RAC) give similar slope independent of the substrate (Slope(RAC) = 0.7 ± 0.1 , Slope(Avicel) = 0.8 ± 0.1). RAC has little or no crystallinity and more than half of the cellulose in Avicel is crystalline (26). Thus, the result in figure S7 also indicate that the scaling can be transferred to other types of cellulosic substrates even though their surfaces differs.

Figure S7. Correlation plot between MM-parameters derived for the enzymes shown in table S2 on amorphous (RAC) and semicrystalline (Avicel) substrate. Solid lines are from the linear regression to the experimental data. Open symbols are kinetic parameters for the two EGs (HiCel45A and TrCel7B) on RAC. These points were omitted from the linear regression analysis.

c) If there is a proportional change in k_{cat} and K_M for the different enzymes (relative to the reference enzyme), the important ratio of k_{cat}/K_M cannot change much. I was wondering what this can mean? I should be quite surprising if all enzymes had the same/similar efficiency or were lacking specificity compared one to another.

One way to interpret the specificity constant (k_{cat}/K_M) is to regard it as a lower limit of the bimolecular rate constant for the complexation (k_{on}). In the simple model used in this study, this would imply that the change in activation free energy of the complexation ($\Delta\Delta G_{on}^\ddagger$) may be estimated as in eq. S12;

$$\Delta\Delta G_{on}^\ddagger = -RT \ln\left(\frac{\varepsilon}{\varepsilon_{ref}}\right), \quad \varepsilon = \frac{k_{cat}}{K_M} \quad (\text{Eq. S12})$$

Where ε_{ref} is the specificity constant of a reference enzyme. By inferring the scaling relationship in eq. S12 we may predict the change in the activation free energy of complexation ($\Delta\Delta G_{on}^\ddagger$) as shown in eq. S13:

$$\Delta\Delta G_{on}^\ddagger = -RT \ln\left(\frac{k_{cat}}{k_{cat,ref}}\right) + RT \ln\left(\frac{K_M}{K_{M,ref}}\right) = \Delta\Delta G^\ddagger + \Delta\Delta G_B^o = \Delta\Delta G_B^o(\Phi + 1) \quad (\text{Eq. S13})$$

Where Φ is the slope in fig. 2 (main text). Equation S13 shows that the change in the activation free energy of complexation ($\Delta\Delta G_{on}^\ddagger$) depends on the binding energy but unlike the activation energy for the overall reaction ($\Delta\Delta G^\ddagger$) the scaling is positive and small since $\Phi + 1 = -0.74 + 1 = 0.26$. To test this prediction, we calculated $\Delta\Delta G_{on}^\ddagger$ for all the investigated enzymes using eq. S13 and plotted these values

as a function of $\Delta\Delta G_B^0$. The results are shown in figure S8 (red symbols). From linear regression we found a slope of 0.24 ± 0.02 which is in excellent accordance with the predicted value. For comparison, we also plotted the change activation energy of threading from the independent kinetic measurements shown in figure S6. Although the two activation energies derived for the complexation differ, it indicates that the change in binding energy have little influence on the free energy of complexation. One interpretation of this behavior is that the transition state (TS) of complexation has similar energy for many cellulases. This conclusion is the same as the one found by Røjel et al. (8) where the authors suggested that the TS lie very early in the reaction path before most of the favorable interactions are made with the substrate. For that reason the difference in binding energy of the different cellulases might only give minor changes in the activation energy of complexation. The overall consequence would be that the specificity constants are not significantly different for the broad range of investigated cellulases, which might also be seen from table S2 (SI, main text).

Figure S8. Correlation plot of change in the binding free energy ($\Delta\Delta G^0_B$, Eq. 2) and activation free energy of complexation ($\Delta\Delta G_{on}^\ddagger$) and dissociation ($\Delta\Delta G_{off}^\ddagger$). The $\Delta\Delta G_{off}^\ddagger$ values are taken from fig. 2 in the main text while the $\Delta\Delta G_{on}^\ddagger$ values are calculated using eq. S13. Blue symbols show $\Delta\Delta G_{on}^\ddagger$ values derived from threading experiments (see fig. S6). Solid lines are from linear regression to the experimental data. The Numbers in the plot indicate the slope of the two regression lines.

2) Considering the limitations pointed out, I find the discussion extremely speculative and for the most part insufficiently supported by the relevant evidence. For example, it is not clear what a fitness landscape for the cellulases should be (Figure 2 and 4). The discussion about uniform/differential binding or selective transition state or ground state binding also seems to lack a proper basis (Figure 5). It can be noted that the original papers on these effects in enzyme catalysis have been very rigorous and clear about the underlying enzymatic-kinetic mechanism. Further transformations (mathematical differentiation) of the simple hyperbola lead to correlations like that in Figure 6 which seem to indicate that enzymes of different K_M differ in catalytic rate depending on the substrate concentration used. I may misunderstand something, but the conclusion seems trivial to me.

In writing, the Abstract can represent various parts in the manuscript that are unclear because of overinterpretation. What are basic physical restrictions on the hydrolytic process? How has evolution “condensed” cellulases to operate along the line of the LFER? What is the link from heterogeneous inorganic catalysis to interfacial enzyme catalysis? I can follow the idea that prediction of the k_{cat} from a computed binding energy has appeal. However, to be relevant in practice, one would have to show that this applies to substrates different from the model cellulose used in their study.

In this study we have discussed the enzyme fitness landscape using the definition by Romero and Arnold (27) where they define the fitness landscape as “The mapping from genotype (target sequence) to phenotype”. Both figure 2 and 4 of the main text show how changes in the genotype (protein sequence in this case) translates into a restricted change in the kinetic parameters (phenotype). We claim that cellulase kinetics is limited by a scaling relations which they cannot escape. We hypothesize that this scaling is due to the nature of the interaction between the substrate and the enzyme, which reduces the complicated reaction mechanism to a binding kinetic problem where k_{cat} and K_M become interlinked. This is a physical restriction that originates from the insoluble nature of the substrate, where the enzyme need to provide work to detach the polymer strand from the semi-crystalline matrix before it can attack the glycosidic bond. For bulk enzymes, the diffusion rate sets an upper limit (physical restriction) for the maximal turnover frequency, but this restriction is rarely observed since most enzymes operates at much slower rates. Cellulases seems to have a more severe physical restriction, which condenses the fitness space to a narrow range defined by the enzyme-substrate affinity. In the discussion by Warshel (28) about different classes of mutations and their effects on the energy diagram, he explicitly uses k_{cat} and K_M in the same way as done here. Based on the answers in the previous sections, we believe that we have justified the use of a MM-based approach to cellulase kinetics and that the parameters may be interpreted in a similar way as was done in the work by Warshel.

Regarding the results shown in figure 6. A direct consequence of the reported scaling relationship is that an optimal affinity exists for a given substrate load. This is a clear illustration of the Sabatier principle for heterogeneous biocatalysis and this provides a direct link to inorganic heterogeneous catalysis where the principle is illustrated by such volcano curves (29). The ability to predict the optimal affinity and hence maximal activity at a given substrate loads (eq. 7, main text) does not seem trivial to us. This could be used in combination with computational prediction of enzyme-substrate binding affinity to screen (in silico) for the optimal cellulases for a given substrate load. In our current response, we have also shown that the scaling is not limited to Avicel (used in the main text) but also extends to a different one, amorphous cellulose (RAC). Further studies are needed to explore the scaling relations for more complex and industrial relevant cellulosic substrates, but this is a work in progress.

Reviewer #3 (Remarks to the Author):

Kari et al submitted a very interesting paper that reports the discovery of a clear lfer among activation and binding free energy for a series of wt and engineered cellulases. The authors propose a new method that uses the lfer for optimizing cellulases and enzymes in general.

The paper is very clear and well written and the results are well supported by the data. The subject is of broad interest for the community even though it's broad applicability is still not convincingly justified (see below). The article is very original and may inspire others to follow the author's path for enzyme optimization.

I recommend publication after the questions raised below are clarified.

We thank the reviewer for the constructive comments. In the following, we have made a point-by-point response to the comments made by the reviewer. To increase readability of our response, comments are shown in red, while our reply is shown without any change in font or color. We highlighted, added or changed text in the main manuscript or SI in green and *italic*. Original text from the manuscript is shown in *italic*.

General questions

g1. The most important aspect that is missing is a discussion on the transferability of the findings to other enzyme families. From a chemistry point of view, cellulases work all through the same chemical mechanism and cross similar transitions states. The nuances are very minor. Do the authors expect the conclusions to other enzyme families? And across enzyme families? The former is crucial to measure the impact of the work. The latter is not so important for enzyme optimization but is interesting per se.

We thank the reviewer for the question. This is a very good point. In this study, we deliberately selected the investigated cellulases so that they covered a broad range of GH families (GH5, GH6, GH7, GH12, GH45), catalytic mechanisms (retaining/inverting) and modes of action (endo-active/exo-active). The idea was to see if the scaling relations were independent of the structural and mechanistic diversity across different GH families.

Cellulases have a catalytic mechanism similar to many other hydrolases. The main difference is that unlike enzymes in the bulk, cellulases work at the solid-liquid interface and need to decrystallize a cellulose strand from the (semi-)crystalline substrate before it can hydrolyse it. This process seems to be slower than the actual catalytic event, since the observed maximal steady-state rate is an order of magnitude smaller than the hydrolysis of the glyosidic bond. For this reason, we do not think that the conclusions made in this study are easily transferred to homogenous biocatalysis. However, other interfacial enzymes may face a similar problem as the cellulases and be limited by the binding kinetics. Since interfacial enzyme reaction are common in both nature and industrial application of enzymes, it seems to be of broad interest to explore whether scaling relations are as common in heterogeneous biocatalysis as it has been found in inorganic heterogeneous catalysis.

To emphasize this point we have now added the following text in the discussion (page 9)

Hence, strong ligand binding appears to benefit catalysis by promoting ligand transfer (48), but it is inevitably associated with a slow turnover of an off-rate controlled reaction, as illustrated in Fig. 5B-3.

We suggest that the observed LFER is a direct consequence of the overall reaction being controlled by the binding kinetics of the cellulases (see figure S6, SI). The existence of LFERs for enzyme reactions governed by the chemical step remains to be investigated, but meta-analyses of kinetic databases show little correlation between k_{cat} and K_M (33,49). Kinetic parameters for heterogeneous enzyme reactions are scarce. Thus, it is still an open question whether scaling relations are as common in heterogeneous biocatalysis as it is in inorganic heterogeneous catalysis (50).

g2. If the kinetics is controlled by product release it is expectable that changes in binding free energy and activation free energy are correlated. In fact, the observed activation free energy may be a surrogate of the binding free energy of the product, which in turn is influenced by the binding free energy of the reactant.

A much more complex case emerges in enzyme families where the observed activation free energy comes from the chemical step.

The authors consider that the LFER might hold in that case? Why?

In both homogenous and heterogeneous (non-biochemical) catalysis LFER are observed even though the activation free energy comes from the chemical step. In many of these cases the catalyst form a covalent intermediate with an atom in the reactant. This competition between the chemical bond offered by respectively the reactant and catalyst intermediate give rise to a trade-off between binding energy (stability of the intermediate) and activation energy (barrier of the intermediate). The enzyme-substrate binding affinity of many enzymes typically originate from the sum of several interactions with the substrate, many of which are not directly involved in the catalytic mechanism. This three dimensional binding environments might be a strategy to decouple the binding energy of the enzyme-substrate intermediate from the activation energy of the chemical step and thus escape the scaling relationship as recently discussed by Vojvodic and Nørskov (30).

Minor points

m1. The discussion at lines 168-176 is a bit confusing. The authors divide fig. 2A into two regions - above and below the diagonal, with the first corresponding to low turnover and weak binding and the second to the opposite. However, the area above the diagonal includes regions of high turnover (SE) and regions of high affinity (NW). What it doesn't include is a region where both are favorable at the same time (i.e., the SW corner). The same comment (in the opposite direction) holds for the region below the diagonal.

We agree that this is confusing. To clarify we have now made the following change in the discussion (page 7).

The area above the diagonal in Fig. 2 represent a region where the enzymes have a low specificity constant ~~combination of low turnover and weak ligand binding~~, and this seems to signify inefficient catalyst. We found some enzymes in this range, including some wild type enzymes and variants with replacements of key amino acid residues. We suggest that this northeastern region of the fitness landscape represents enzymes that have been either catalytically impaired by our engineering, ~~are structurally unstable under the selected conditions~~ ~~structurally unstable~~, or have other primary substrate preference than cellulose.

On the other hand, the region below the diagonal in Fig 2, specifies enzymes which have a high specificity constant ~~have both strong substrate binding and rapid turnover (both $\frac{-\Delta\Delta G^o}{B}$ and $-\Delta\Delta G^\ddagger$ are low compared to the reference).~~ These traits clearly appear functionally advantageous....

m2. Concerning the physical origin of the correlation, the authors propose that it reflects an equilibrium between evolutionary selection and physical constraints.

This would mean that evolution drives the enzymes always in the way toward higher rates. At ref. 26 (and in an upcoming paper including one thousand enzymes from all enzyme classes) we have seen that enzyme efficiency is narrowed to a sharp range, which suggests that enzymes may evolve not to work as fast as possible but instead at the right speed that maximizes overall fitness and the "concerted metabolism" of the organism. An excess of activity may break down homeostasis. Also, if we can evolve enzymes to make them to work faster this might mean that the physical constraints are not insurmountable. What the authors think about this?

Well, all of this are educated guesses, we don't really know the answers for these fascinating questions and I perfectly respect the opinion of the authors.

Again, we thank the reviewer for an insightful question. In this study we propose that cellulases have severe physical restraints given by the LFER and that this have the direct consequence that the best cellulase (optimal activity) dependent of the experimental conditions. Hence, temperature and substrate accessibility might shift the optimal balance between affinity and activity as shown in figure 6 (main text). In nature cellulases may have optimized their binding affinity to the environmental conditions but they may also, as the reviewer suggest, have evolved to escape the LFER. For instance, all fungi that secrete cellulase extracellularly produce a cocktail of cellulases, which are known to synergize. We are currently investigating whether cellulase synergy serve to escape the LFER but further studies will be needed. Another important aspect which may have affected the evolution of cellulases is the uptake of glucose by the organism. Since cellulases essentially produce glucose, a high production rate would not necessary be favorable for the organism if it cannot import the molecule at the same speed as it produce it.

References

1. Lemkul, J. A., and Bevan, D. R. (2010) Assessing the Stability of Alzheimer's Amyloid Protofibrils Using Molecular Dynamics. *The Journal of Physical Chemistry B* **114**, 1652-1660
2. Payne, C. M., Jiang, W., Shirts, M. R., Himmel, M. E., Crowley, M. F., and Beckham, G. T. (2013) Glycoside hydrolase processivity is directly related to oligosaccharide binding free energy. *J Am Chem Soc* **135**, 18831-18839
3. Colussi, F., Sorensen, T. H., Alasepp, K., Kari, J., Cruys-Bagger, N., Windahl, M. S., Olsen, J. P., Borch, K., and Westh, P. (2015) Probing substrate interactions in the active tunnel of a catalytically deficient cellobiohydrolase (Cel7). *The Journal of biological chemistry* **290**, 2444-2454
4. Medford, A. J., Vojvodic, A., Hummelshøj, J. S., Voss, J., Abild-Pedersen, F., Studt, F., Bligaard, T., Nilsson, A., and Nørskov, J. K. (2015) From the Sabatier principle to a predictive theory of transition-metal heterogeneous catalysis. *Journal of Catalysis* **328**, 36-42

5. Chundawat, S. P. S., Beckham, G. T., Himmel, M. E., and Dale, B. E. (2011) Deconstruction of lignocellulosic biomass to fuels and chemicals. *Annu Rev Chem Biomol* **2**, 121-145
6. Payne, C. M., Knott, B. C., Mayes, H. B., Hansson, H., Himmel, M. E., Sandgren, M., Stahlberg, J., and Beckham, G. T. (2015) Fungal cellulases. *Chem Rev* **115**, 1308-1448
7. Cruys-Bagger, N., Alasepp, K., Andersen, M., Ottesen, J., Borch, K., and Westh, P. (2016) Rate of Threading a Cellulose Chain into the Binding Tunnel of a Cellulase. *Journal of Physical Chemistry B* **120**, 5591-5600
8. Røjel, N., Kari, J., Sørensen, T. H., Badino, S. F., Morth, J. P., Schaller, K., Cavaleiro, A. M., Borch, K., and Westh, P. (2019) Substrate binding in the processive cellulase Cel7A: Transition state of complexation and roles of conserved tryptophan residues.
9. Chundawat, S. P., Bellesia, G., Uppugundla, N., da Costa Sousa, L., Gao, D., Cheh, A. M., Agarwal, U. P., Bianchetti, C. M., Phillips, G. N., Jr., Langan, P., Balan, V., Gnanakaran, S., and Dale, B. E. (2011) Restructuring the crystalline cellulose hydrogen bond network enhances its depolymerization rate. *J Am Chem Soc* **133**, 11163-11174
10. Igarashi, K., Uchihashi, T., Koivula, A., Wada, M., Kimura, S., Okamoto, T., Penttila, M., Ando, T., and Samejima, M. (2011) Traffic jams reduce hydrolytic efficiency of cellulase on cellulose surface. *Science* **333**, 1279-1282
11. Cruys-Bagger, N., Tatsumi, H., Ren, G. R., Borch, K., and Westh, P. (2013) Transient kinetics and rate-limiting steps for the processive cellobiohydrolase Cel7A: effects of substrate structure and carbohydrate binding domain. *Biochemistry* **52**, 8938-8948
12. Cruys-Bagger, N., Elmerdahl, J., Praestgaard, E., Tatsumi, H., Spodsberg, N., Borch, K., and Westh, P. (2012) Pre-steady-state kinetics for hydrolysis of insoluble cellulose by cellobiohydrolase Cel7A. *The Journal of biological chemistry* **287**, 18451-18458
13. Knott, B. C., Haddad Momeni, M., Crowley, M. F., Mackenzie, L. F., Götz, A. W., Sandgren, M., Withers, S. G., Ståhlberg, J., and Beckham, G. T. (2014) The Mechanism of Cellulose Hydrolysis by a Two-Step, Retaining Cellobiohydrolase Elucidated by Structural and Transition Path Sampling Studies. *Journal of the American Chemical Society* **136**, 321-329
14. Fersht, A. (1985) *Enzyme structure and mechanism*, 2 ed., W.H. Freeman, San Francisco
15. Kurasin, M., and Valjamae, P. (2011) Processivity of cellobiohydrolases is limited by the substrate. *The Journal of biological chemistry* **286**, 169-177
16. Murphy, L., Cruys-Bagger, N., Damgaard, H. D., Baumann, M. J., Olsen, S. N., Borch, K., Lassen, S. F., Sweeney, M., Tatsumi, H., and Westh, P. (2012) Origin of initial burst in activity for *Trichoderma reesei* endo-glucanases hydrolyzing insoluble cellulose. *The Journal of biological chemistry* **287**, 1252-1260
17. Christensen, S. J., Kari, J., Badino, S. F., Borch, K., and Westh, P. (2018) Rate-limiting step and substrate accessibility of cellobiohydrolase Cel6A from *Trichoderma reesei*. *The FEBS journal* **285**, 4482-4493
18. Cruys-Bagger, N., Elmerdahl, J., Praestgaard, E., Tatsumi, H., Spodsberg, N., Borch, K., and Westh, P. (2012) Pre-steady state kinetics for the hydrolysis of insoluble cellulose by cellobiohydrolase Cel7A. *The Journal of biological chemistry* **287**
19. Igarashi, K., Koivula, A., Wada, M., Kimura, S., Penttila, M., and Samejima, M. (2009) High speed atomic force microscopy visualizes processive movement of *Trichoderma reesei* cellobiohydrolase I on crystalline cellulose. *The Journal of biological chemistry* **284**, 36186-36190
20. Kari, J., Kont, R., Borch, K., Buskov, S., Olsen, J. P., Cruys-Bagger, N., Valjamae, P., and Westh, P. (2017) Anomeric Selectivity and Product Profile of a Processive Cellulase. *Biochemistry* **56**, 167-178

21. Fromm, S. J., and Fromm, H. J. (1999) A Two-Step Computer-Assisted Method for Deriving Steady-State Rate Equations. *Biochemical and Biophysical Research Communications* **265**, 448-452
22. Praestgaard, E., Elmerdahl, J., Murphy, L., Nymand, S., McFarland, K. C., Borch, K., and Westh, P. (2011) A kinetic model for the burst phase of processive cellulases. *The FEBS journal* **278**, 1547-1560
23. Kari, J., Olsen, J., Borch, K., Cruys-Bagger, N., Jensen, K., and Westh, P. (2014) Kinetics of Cellobiohydrolase (Cel7A) Variants with Lowered Substrate Affinity. *The Journal of biological chemistry* **289**, 32459-32468
24. Bar-Even, A., Noor, E., Savir, Y., Liebermeister, W., Davidi, D., Tawfik, D. S., and Milo, R. (2011) The Moderately Efficient Enzyme: Evolutionary and Physicochemical Trends Shaping Enzyme Parameters. *Biochemistry* **50**, 4402-4410
25. Sousa, S. F., Ramos, M. J., Lim, C., and Fernandes, P. A. (2015) Relationship between Enzyme/Substrate Properties and Enzyme Efficiency in Hydrolases. *ACS Catalysis* **5**, 5877-5887
26. Zhang, Y. H., and Lynd, L. R. (2004) Toward an aggregated understanding of enzymatic hydrolysis of cellulose: noncomplexed cellulase systems. *Biotechnology and bioengineering* **88**, 797-824
27. Romero, P. A., and Arnold, F. H. (2009) Exploring protein fitness landscapes by directed evolution. *Nat Rev Mol Cell Bio* **10**, 866-876
28. Warshel, A. (1998) Electrostatic Origin of the Catalytic Power of Enzymes and the Role of Preorganized Active Sites. *J. Biol. Chem.* **273**, 27035-27038
29. Bligaard, T., Nørskov, J. K., Dahl, S., Matthiesen, J., Christensen, C. H., and Sehested, J. (2004) The Brønsted–Evans–Polanyi relation and the volcano curve in heterogeneous catalysis. *Journal of Catalysis* **224**, 206-217
30. Vojvodic, A., and Nørskov, J. K. (2015) New design paradigm for heterogeneous catalysts. *National Science Review* **2**, 140-143

REVIEWER COMMENTS

Reviewer #1 (Remarks to the Author):

The authors have addressed my concerns fully. The paper should be published now.

Reviewer #2 (Remarks to the Author):

(Please refer to file on the following page)

Reviewer #3 (Remarks to the Author):

The authors have made a tremendous work answering the questions posed by the three referees. In general, I consider that the answers have addressed my concerns, and in my opinion, the article is ready for publication.

I note that a study similar to the meta-analysis referred to in ref. 33 is now available in the same journal, but using a much vaster database, which provides more solid evidence for the non-triviality of the existence of a correlation between activation and binding free energies. As a final note, I also suggest the authors include a brief paragraph in the final manuscript with the answer for question g2, as I believe this question will come to the mind to many readers.

Pedro Alexandrino Fernandes

Reviewer #2:

The authors have responded to my points. To do so, they justified their position and provided new experimental data from kinetic studies of enzyme binding to substrate. I appreciate their efforts. However, my concerns remain.

To avoid misunderstanding, I wish to make clear that my criticism is not on the use of the Michaelis-Menten relationship. This fits the data and can also be practical in various ways. My criticism is on the interpretation of K_M as a parameter of binding and a set of further conclusions resting on that interpretation. I will explain below why this is false in the relevant context. I also wish to make clear that I haven't implied a universal correlation between K_M and k_{cat} among enzymes (page 10 of the rebuttal). But even the reference cited in the rebuttal letter (Bar-Even et al. #24) states that "More specifically, the global data indicate only a minor correlation between k_{cat} and K_M and thus fail to indicate a clear trade-off between these parameters. (*My emphasis is in italics!!*) However, analysis of enzymatic reactions with similar properties might reveal that this trade-off prevails in numerous cases (e.g., ref 52)." I would argue that the hydrolysis of Avicel cellulose is a clear case of an "enzymatic reaction with similar properties". So, it seems important to be cautious about any "trade-off" that might be kinetically caused.

1) The model used in Figure S1 to describe the enzymatic cycle is not clear. It involves an adsorbed enzyme that undergoes "threading" (rate constants k_2 and k_{-2}) to load a cellulose chain into the binding pocket. During dissociation, the enzyme must become "unthreaded" from the chain and is then released from the surface. Now, while "threading" is described as a reversible process, the dissociation including "unthreading" is an irreversible process. It is not clear why. The "unthreading" step of the binding process is shown to occur with the same rate constant (k_{-2}) as the dissociation step. The model appears to violate the principle of Occam's razor: why are there separate complexes for "threading" and for "unthreading" leading to dissociation post-catalysis, if it is assumed that the involved steps occur at exactly the same rate? A parsimonious model description would go back to the "threading complex" from which the enzyme can dissociate anyway. Conversely, if there are unique complexes for binding and dissociation, how can it be justified that the steps occur with the same rate constant? One cannot have both, unless there is compelling evidence, which is however not provided.

It is instructive to consider a model in which the dissociation is assigned a different rate constant, here referred to as k_4 . I show below that with the assumption that k_{-2} and k_4 both are slow steps of the kinetic mechanism, the rate constant k_{-2} disappears from the expression of K_M . In a kinetic mechanism that involves rate-limiting dissociation of the enzyme, the K_M can be largely determined by k_4 . Moreover, the whole notion of K_M as a binding constant collapses.

I also find it useful to consider a previous paper from the same group of authors (JBC 2014, 289, 32489) in which variants of Cel7A with lowered substrate affinity were analyzed. In this paper, relationships of K_M ($= k_{off}/k_{on}$; Eq. 2) and the maximum rate V_{max} are given. (The V_{max} is the equivalent of the k_{cat} used in their current manuscript.) Eq. 5 in the 2014 JBC paper shows that the V_{max} (k_{cat}) is directly proportional to k_{off} ($V_{max} \approx n k_{off} E_0$) where E_0 is the enzyme concentration and n is the number of processive steps. It becomes evident immediately that k_{cat} and K_M are not independent parameters and (linear) correlation between them is actually as expected from the kinetic relationships. It is discussed in the

2014 JBC paper that affinity for substrate goes down (K_M increases) when k_{cat} goes up. It is important to see the point: what is presented as an insightful LFER between a binding parameter and a rate parameter can largely become a correlation of one parameter with effectively itself.

The relationship between the apparent free energies calculated from the experimental parameters and the energy barriers in Figure 5 is not clear. The scheme in Figure 5A is not the one used in the kinetic analysis (Scheme S1).

2) I found the analysis of enzyme binding unclear. First of all, how can a two-step binding model (Scheme S1) be reduced to a single-step binding model (Scheme S2), solely on account of the fact that there is a spectroscopic reporter for the bound (“threaded”) enzyme. Secondly, and quite importantly, what makes them believe that the data obtained are just for binding alone? The bound enzyme will react quickly because chain cleavage is fast and if Scheme S1 is correct, the enzyme complex before dissociation will accumulate. I cannot determine from the data shown whether this complex also has fluorescence; but even if it has not, the dissociation rate (note: k_4 not k_2) will affect the outcome of the binding experiment. Therefore, the K_D is not a binding constant in the way interpreted. Figure S6 shows a correlation as it must be because k_{on} , k_{off} and K_D are not independent one from another.

Appendix (kinetic model)

Proposed mechanism in the manuscript

Consider that k_{+4} replaces k_2 to described the last step (dissociation of the enzyme)

$$k_{cat} = \frac{nk_{+2}k_{+3}k_{+4}}{k_{+4}(k_{-2} + k_{+3}) + k_{+2}(k_{+4} + k_{+3})}$$

... if $k_{+3} \gg k_{-2}$ and k_{+4}

$$k_{cat} = \frac{nk_{+2}k_{+4}}{k_{+4} + k_{+2}} \cong nk_{+4}$$

$$K_m = \frac{k_{+4}(k_{-1}(k_{-2} + k_{+3}) + k_{+2}k_{+3})}{k_{+1}(k_{+4}(k_{-2} + k_{+3}) + k_{+2}(k_{+4} + k_{+3}))}$$

... if $k_{+3} \gg k_{-2}$ and k_{+4}

$$K_m = \frac{k_{+4}(k_{-1} + k_{+2})}{k_{+1}(k_{+4} + k_{+2})}$$

Response to reviewer 2

We would like to thank the reviewer for the persistent and constructive feedback. This type of discussion is rarely seen anymore and we appreciate the time used by the reviewer to engage in the scientific discussion. The concerns raised by the reviewer revolve around the meaning/interpretation of the reported apparent Michaelis-Menten constants (K_M) and the origin of the correlation between the apparent MM-parameters (K_M and k_{cat}). Below is a point-by-point answer to the questions and concerns raised by the reviewer. However, before going into a detailed response we would like to make some general comments to the critique (see below).

The original comments from the referees are shown in red, while our reply is shown without any change in font or color.

General comments

In the first response letter, we introduced a kinetic model (in the following referred to as model 1) with several steps to show that even complex reaction schemes may be simplified to a MM-equation. Further, the idea was to show that the scaling between the apparent K_M and k_{cat} only arises when the same rate-constant acts as a scaling parameter in both the expression for K_M and k_{cat} . Model 1 was introduced as a response to the critique given in the first response letter but it was not intended as a general model for cellulolytic enzymes or as a general framework to guide our interpretation of the apparent MM-parameters. We will like to emphasize this point to avoid a detailed discussion about model 1 rather than a discussion of the main work in the article. We are aware that a steady-state model and its (apparent) parameters are entangled since a true understanding of the MM-parameters will require a complete molecular description of the overall reaction. However, for cellulases, and most heterogeneous enzyme reactions, we are very far from having a detailed molecular theory and further from a kinetic model with experimentally determined rate constants for all elementary steps. For the best-studied cellulase (reducing-end-specific cellobiohydrolase from *Trichoderma reesei*), around a dozen elementary reactions have been proposed but less than half of these steps have been quantified experimentally. For other types of cellulases belonging to different families and with other modes of action, essentially no information exists about their overall reaction mechanism.

The approach here is an attempt to do comparative kinetic analysis despite the complex nature of the substrate and structural diversity of the investigated enzymes. In our opinion, the observed correlation between K_M and k_{cat} interconnects the kinetics of this diverse group of cellulases and simplifies the otherwise inaccessible and complex kinetic analysis. In doing so, it also greatly reduces the complexity of the parameter space to essentially a one-dimension space given by the apparent Michaelis-Menten constant (K_M). Perspectives of this approach are particularly promising inasmuch as it has been successfully used within inorganic heterogeneous catalysis. For instance, it was found that the hydrogenation of CO to form ethanol, a process that consists of 52 elementary steps, could be described by essential two descriptors (the binding strength of the inorganic catalyst to carbon and oxygen) (1). Since all other steps scaled with these descriptors, only two parameters were needed to characterize different catalysts for this complicated reaction. Finding such operational descriptors tremendously reduced the complexity of the kinetic analysis and convert an otherwise multi-dimensional parameter space to a few experimentally accessible parameters.

Whether the apparent K_M value is an operational affinity parameter or a true dissociation constant for the enzyme-substrate complex will depend on the molecular model underlying each class of the investigated

cellulases. This distinction, however, is not critical in the current context. The conspicuous correlation between enzyme-substrate affinity and activity across a diverse group of cellulolytic enzymes is novel and useful for both analyses and design of cellulolytic enzyme, as indeed noted by the other reviewers. The apparent Michaelis-Menten constant is by definition the substrate load at which half of the enzymes are complexed (productively) with the substrate. This is indisputable a descriptor of the stability of the productive enzyme-substrate complex. In our opinion, the main critique is semantic as it is a matter of K_M being interpreted as a descriptor of affinity or equilibrium constant.

In this study, we characterized 83 cellulases covering 6 different glycosyl hydrolase families (See table 1, main text) and provided evidence for a direct correlation between the apparent K_M and equilibrium constant for the enzyme-substrate complex derived from independent measurements. In the lack of a detailed and adequate kinetic model, we believe that the discussion of the apparent Michaelis-Menten constant and the origin of the scaling should be rooted in such measurements.

Specific questions

In the following, we will reply to each of the questions and concerns. However, we will like to stress that a detailed discussion of model 1 is beyond the scope of the main text since model 1 was not introduced as a general model for cellulase kinetics (see “General comments” above).

The model used in Figure S1 to describe the enzymatic cycle is not clear. It involves an adsorbed enzyme that undergoes “threading” (rate constants k_2 and k_2) to load a cellulose chain into the binding pocket. During dissociation, the enzyme must become “unthreaded” from the chain and is then released from the surface. Now, while “threading” is described as a reversible process, the dissociation including “unthreading” is an irreversible process. It is not clear why.

The controversy arises because S_i is not the same as S_{i-n} . S_{i-n} is a dead-end at which the enzyme cannot progress catalysis further. Once a productive enzyme-substrate complex has been formed the enzyme has reduced its translational freedom to one dimension. This makes the enzyme vulnerable to surface obstacles as it can only escape the complex by dissociation if its path is blocked (See figure S1). By contrast, at any earlier position in the processive run, the enzyme can either progress or dissociate.

Figure S1. Illustration of dissociation from an unproductive enzyme-substrate complex (ES_{i-n}).

The model permits binding to S_{i-n} but the concentration of these unproductive sites will be small compared to S_i in the initial part of the reaction. Under the experimental condition used in this study, we can neglect the backward reaction since the concentration of S_{i-n} is small. We also neglect product inhibition (product unbind irreversibly) with similar argumentation although the product can bind and unbind reversible to the product site of the enzyme. The validity of this approach has been investigated previously (2) but generally, the initial rate measurements and the low enzyme to substrate ratio ($E_0 \ll S_0$) allow us to neglect product inhibition, physical instability of the enzyme, binding to S_{i-n} and substrate depletion.

The “unthreading” step of the binding process is shown to occur with the same rate constant (k_2) as the dissociation step. The model appears to violate the principle of Occam’s razor: why are there separate

complexes for “threading” and for “unthreading” leading to dissociation post-catalysis, if it is assumed that the involved steps occur at exactly the same rate? A parsimonious model description would go back to the “threading complex” from which the enzyme can dissociate anyway. Conversely, if there are unique complexes for binding and dissociation, how can it be justified that the steps occur with the same rate constant? One cannot have both, unless there is compelling evidence, which is however not provided. It is instructive to consider a model in which the dissociation is assigned a different rate constant, here referred to as k_4 . I show below that with the assumption that k_{-2} and k_4 both are slow steps of the kinetic mechanism, the rate constant k_{-2} disappears from the expression of K_M . In a kinetic mechanism that involves rate-limiting dissociation of the enzyme, the K_M can be largely determined by k_4 . Moreover, the whole notion of K_M as a binding constant collapses.

Before addressing this (valid) criticism of model 1, we again emphasize that model 1 was introduced to illustrate a point in the first response letter, and that it is peripheral to the main conclusions of this work.

It is correct that the dissociation event after catalysis is simplified to one step instead of two. The explanation is connected to the previous answer since it is due to the assumed irreversibility of dissociation from a “dead-end” (dissociation post-catalysis). In figure S2, a two-step dissociation scheme from a dead-end is illustrated. Both dethreading and desorption are assumed to be irreversible in the initial part of the reaction where $S_1 \gg S_{1-n}$.

Figure S2. Illustration of a two-step dissociation from an unproductive enzyme-substrate complex (ES_{i-n}).

At steady-state, any sequence of first-order reactions will be controlled by the slowest step in the forward reaction path. As discussed in the first response letter the dethreading is much slower than the desorption of a CBM-bound cellulase (3). Hence, the first step in figure S2 controls the overall rate of this two-step dissociation. This was the main reason why we condensed the dissociation from a dead-end to one step, and why this step was not assigned with a separate rate-constant as proposed by the reviewer (e.g. k_4). The validity of this approach will depend on the magnitude of the rate-constant that governs step 1 and 2 in figure S2. If the second step (middle-step in figure S2) becomes comparable to the first step (dethreading) a separate rate-constant would be needed.

It is not surprising that k_{-2} disappears from the expression of K_M when a separate rate-constant is inferred for the dissociation of the ES_{np} complex. When k_{-2} is much slower than k_3 the ES_{thread} will almost exclusively decay through the catalytic step(s). Under such conditions, the intermediate threaded complex is redundant and the overall reaction may be simplified to the model shown in scheme S1.

This simple model has condensed the binding and unbinding into one step and it is likely to be more general for the type of reaction studied here. For instance, some of the investigated cellulases do not have a CBM module, which obviates the need for a two-step binding.

The model is similar to an *ordered uni-bi reaction* (4) and at steady-state the rate-equation can be expressed as shown in eq. S5. The model also resembles the proposed mechanism illustrated in the main manuscript (figure 5A, main manuscript). The steady-state rate-equation for scheme S1 is given in eq. S2.

$$v = \frac{E_0 S_0 n k_{off} \frac{k_{cat}}{k_{off} + k_{cat}}}{\frac{k_{off}}{k_{on}} + S_0} = \frac{S_0^{app} V_{max}}{^{app} K_M + S_0} \quad (\text{eq. S2})$$

For the (few) cellulases that have been kinetically characterized in some detail, $k_{cat} \gg k_{off}$, and the apparent MM-parameter may be expressed as follow

$$\begin{aligned} ^{app} V_{max} &: E_0 n k_{off} \quad , \quad k_{cat} \gg k_{off} \\ ^{app} K_M &= \frac{k_{off}}{k_{on}} \end{aligned} \quad (\text{eq. S3})$$

If separate parameters are given for the dissociation of the ES and ES* complex as suggested by the reviewer (see scheme 2), the steady-state rate-equation can be expressed as in eq. S4.

$$E + S \xrightleftharpoons[k_{off1}]{k_{on}} ES \xrightarrow{n^* k_{cat}} nP + ES^* \xrightarrow{k_{off2}} E + S^* \quad (\text{scheme S2})$$

$$v = \frac{E_0 S_0 n k_{off2} \frac{k_{cat}}{(k_{off2} + k_{cat})}}{\frac{k_{off2}(k_{off1} + k_{cat})}{k_{on}(k_{off2} + k_{cat})} + S_0} = \frac{S_0^{app} V_{max}}{^{app} K_M + S_0} \quad (\text{eq. S4})$$

If we again assume that $k_{cat} \gg k_{off1}$ and $k_{cat} \gg k_{off2}$ the apparent MM-parameter may now be expressed in the following way.

$$\begin{aligned} ^{app} V_{max} &: E_0 n k_{off2} \quad , \quad k_{cat} \gg k_{off2} \\ ^{app} K_M &= \frac{k_{off2}(k_{off1} + k_{cat})}{k_{on}(k_{off2} + k_{cat})} : \frac{k_{off2}}{k_{on}} \quad , \quad k_{cat} \gg k_{off1} \wedge k_{off2} \end{aligned} \quad (\text{eq. S5})$$

Note that the expression for K_M in the case of two rate-constant (eq. S5) is similar to the expression given by the reviewer in the appendix, with the only difference that k_{cat} is used instead of k_2 .

From the simple model it is clear that the apparent K_M can be understood as a dissociation constant if $k_{cat} \gg k_{off}$. The assumption that $k_{cat} \gg k_{off} \wedge k_{off2}$ may be justified from pre-steady-state measurements. If the rate-limiting step at steady-state is located after the release of product an initial burst in activity is expected since there will be a build-up of unproductive enzymes (ES*) in the pre-steady-state phase. Burst kinetics has been observed for all the types of enzymes studied here except for cellulases from the GH45 family (see table S1). To test whether GH45 also exhibits burst kinetics (this is unstudied hitherto) we made new experiment with real-time amperometric biosensor measurements of the soluble product of HiCel45A. The biosensor was made as reported previously (5) and the substrate load and enzyme concentration was 10 g/L and 0.1 μM respectively.

Figure S3. Real-time measurement of soluble products during the first 250 seconds of the reaction with $0.1\mu\text{M}$ HiCel45A and 10 g/L Avicel. Upper panel show the progress curves while the lower curve show the rate (derivate of the progress curve). The dotted line show the best-fit linear regression to the near linear steady-state region from 200s to 250s. The measurement was done at $25\text{ }^\circ\text{C}$.

As seen from fig. S3, HiCel45A show an initial burst followed by a slower steady-state rate. Linear regression to the near-linear part of the progress curve (200s \rightarrow 250s) shows a steady-state rate of $0.011\text{ }\mu\text{M/s}$. Using the reported steady-state parameters from the supplementary material in the main article and the MM-equation we can compare this rate with the MM-predictions. The K_M and k_{cat} for HiCel45A is 97 g/L and 1.27 s^{-1} . Using an enzyme concentration of $0.1\text{ }\mu\text{M}$ and a substrate load of 10 g/L this give a predicted steady-state rate of:

$$v_{ss} = \frac{E_0 k_{cat} S_0}{K_M + S_0} = \frac{0.1\mu\text{M} \cdot 1.27\text{ s}^{-1} \cdot 10\frac{\text{g}}{\text{L}}}{97\frac{\text{g}}{\text{L}} + 10\frac{\text{g}}{\text{L}}} = 0.012\frac{\mu\text{M}}{\text{s}}$$

This number is in excellent agreement with the steady-state rate from the biosensor measurements ($0.011\text{ }\mu\text{M/s}$).

Table S1. Burst kinetics has been observed experimentally in all of the investigated cellulase groups.

GH family	Structural fold	Catalytic mechanism	Mode of action	EC number	Burst kinetics	ref
7	β -jelly roll	Retaining	CBH	3.2.1.176	Yes (TrCel7A)	(6-11)
			EG	3.2.1.4	Yes (TrCel7B)	(12)
6	α/β barrel	Inverting	CBH	3.2.1.91	Yes (TrCel6A)	(2,13)
5	$(\beta/\alpha)_8$	Retaining	EG	3.2.1.4	Yes (TrCel5A)	(12)
12	β -jelly roll	Retaining	EG	3.2.1.4	Yes (TrCel12A)	(12)
45	β barrel	Inverting	EG	3.2.1.4	Yes (HiCel45A)	Fig. S3

It is difficult to discuss whether the interpretation of the apparent MM parameters are correct/wrong solely based on kinetic models since the answer will depend on the type of model and the magnitude of the rate-constant that governs each step. In the answer above we have shown that a simplified model (scheme S1), which is similar to what we have previously developed for cellulases activity (7,14), the apparent K_M is a true dissociation constant for the enzyme-substrate complex. This type of model also predicts burst kinetics in the pre-steady-state phase which has been found for a representative cellulase of all the investigated cellulase families (See table S1). This supports the view that K_M as measured here is indicative of a dissociation constant, but we acknowledge that the true meaning of the apparent parameter will depend on the true type of model and the magnitude of the rate constants for the individual steps in the model.

I also find it useful to consider a previous paper from the same group of authors (JBC 2014, 289, 32489) in which variants of Cel7A with lowered substrate affinity were analyzed. In this paper, relationships of K_M ($= k_{off}/k_{on}$; Eq. 2) and the maximum rate V_{max} are given. (The V_{max} is the equivalent of the k_{cat} used in their current manuscript.) Eq. 5 in the 2014 JBC paper shows that the V_{max} (k_{cat}) is directly proportional to k_{off} ($V_{max} \approx n k_{off} E_0$) where E_0 is the enzyme concentration and n is the number of processive steps. It becomes evident immediately that k_{cat} and K_M are not independent parameters and (linear) correlation between them is actually as expected from the kinetic relationships. It is discussed in the 2014 JBC paper that affinity for substrate goes down (K_M increases) when k_{cat} goes up. It is important to see the point: what is presented as an insightful LFER between a binding parameter and a rate parameter can largely become a correlation of one parameter with effectively itself. The relationship between the apparent free energies calculated from the experimental parameters and the energy barriers in Figure 5 is not clear. The scheme in Figure 5A is not the one used in the kinetic analysis (Scheme S1).

What is trivial to the reviewer is in our opinion one of the main findings. It is correct that you can construct a model, like the one used in Kari et al. (15) from which it may be deduced that under a given assumption (in this case $k_{off} \ll k_{cat}$) there is a direct correlation between K_M and k_{cat} . Finding such a correlation is a result that supports this type of interpretation since it only holds if the system behaves as hypothesized (i.e. catalysis is much faster than dissociation). In a recent study (16) this interpretation was corroborated for one specific enzyme (Cel7A), but here we show that it is general throughout a large group of cellulases that are structurally and mechanistically diverse. There is no a priori reason to expect a correlation between K_M and k_{cat} . For complex reactions, it greatly simplifies both comparative kinetics and the design of better enzymes. Stating that the relation is trivial based on a model, which is developed to explain the observation is reversing the causality. The 83 independently measured MM parameters show a correlation. The interpretations of the strong correlation rely on both the kinetic model, structural analysis, and previous research concerning the kinetics of cellulases. An experimental supported hypothesis is that cellulases are prone to get stuck in long-lived, unproductive complexes and hence the dissociation rate governs the overall reaction. Individual analysis of both non-reducing end (13) and reducing end specific cellobiohydrolases (6-9,11,15,17) as well as endoglucanases (12) have shown that these enzymes are limited by dissociation. The reviewer state that the correlation between k_{cat} and K_M arises since the same rate-constant appears in both parameters. This is only true if $k_{cat} \gg k_{off}$ and hence we see this as an interpretation of the results. We do not disagree with this interpretation and it is exactly how we explain the correlation in the main manuscript. Figure 5B (main manuscript) serve to illustrate this point.

We believe that the type of analysis presented in figure 5B provides a useful and intuitive illustration of our interpretation on a qualitative level. This is a well established framework for comparative kinetic discussions of isoenzymes (18) and analogous to the so-called Φ -analysis used in protein folding research. In many cases, the protein folding problem has been investigated using a simplified two-state reaction mechanism including only the native (N) and denatured (D) state (19).

From this simplified model of the system it is evident that the equilibrium constant and rate constant for the folding is directly related. Nevertheless, the nature of protein folding reactions has been investigated by comparing the thermodynamic and kinetic parameters of mutants with the corresponding values for the wild type protein (19). Although there is a direct and simple correlation between the equilibrium constant and the rate-constant it does not necessarily mean that an LFER exist between the free energy of folding and the free energy of the transition state. Only if the interactions is present in both the ground state and the TS will a linear relationship exist between the change in the folding energy and activation energy (see figure S4). These considerations have been instrumental to understand the structural nature of the transition state in protein folding research as illustrated in Fig. S4.

Figure S4. Illustration of an energy diagram for a two-state folding model. Upper diagram show the free energy of a wild type protein and the lower diagram show the wild type protein and a variant (red curve) with reduced interaction in both the transition state for the folding and the folded state. As a consequence both the free energy of folding and the free energy of the transition state would scale for the variant. If a group of variants all followed this behavior it would give raise to a LFER where the free energy of folding scaled with the free energy of the transition state.

The analogy to Φ -analysis in protein folding research serve to show that scaling relationship between K_M and k_{cat} does not necessary exist just because the same rate-constant appear as a scaling parameter in both parameters. However, if the two parameters scale (no matter how trivial it may seem) it opens up for in silico screening based on computed substrate affinity.

I found the analysis of enzyme binding unclear. First of all, how can a two-step binding model (Scheme S1) be reduced to a single-step binding model (Scheme S2), solely on account of the fact that there is a spectroscopic reporter for the bound (“threaded”) enzyme. Secondly, and quite importantly, what makes them believe that the data obtained are just for binding alone? The bound enzyme will react quickly because chain cleavage is fast and if Scheme S1 is correct, the enzyme complex before dissociation will accumulate. I cannot determine from the data shown whether this complex also has fluorescence; but even if it has not, the dissociation rate (note: k_4 not k_2) will affect the outcome of the binding experiment. Therefore, the K_D is not a binding constant in the way interpreted. Figure S6 shows a correlation as it must be because k_{on} , k_{off} and K_D are not independent one from another.

The signal from the threading experiments reports dequenching as water is removed from exposed tryptophans upon complexation with a cellulose chain. This is the best operational measure of the population of threaded enzyme (16). We cannot distinguish between productively and unproductively threaded enzymes since these two populations would give rise to the same signal (they both have a ligand in the active site). Hence, the signal reports the sum of these two populations. As a result, the on-rate can be derived in a model free fashion since the initial slope for $t \rightarrow 0$ is a direct measure of the threading rate. We simply measure the rate at which the enzyme go from the free state at $t=0$ to the treaded state. As explained in the supplementary material we can neglect the dethreading (backward reaction) by taking the tangent to the time curves in the very early phase of the reaction (e.g. $t \rightarrow 0$). This threading rate is very similar for the investigated cellulases despite their large difference in both structure, mode of action and steady-state kinetic parameters (apparent MM-parameters). Unlike k_{on} the dissociation constant (K_d) is based on a two-step model. However, the validity of this analysis is supported by experiments with catalytic deficient variants. For TrCel7A it has been found that a catalytic deficient variant had similar K_d and k_{on} (16). This result supports the current interpretation of the threading data which is based on a two-step binding scheme. We have recently measured the Langmuir binding isotherms for TrCel7A and TrCel6A as well as their respective catalytic deficient variants (20). As shown in figure S5 the binding is not significantly affected by the catalysis since the catalytic deficient variants have close to identical binding isotherms. These binding-isotherms further support the idea that the enzyme-substrate complex can not decay by catalysis and that a two-step model is sufficient to describe the complexation reaction.

Figure S4. Binding isotherms for TrCel7A and TrCel6A and two catalytic variants of these two enzymes. The substrate was microcrystalline cellulose (Avicel) and temperature was 25°C. Data taken from (20).

Response to reviewer 3

In the following reviewer comments are shown in red, while our reply is shown without any change in font or color. We highlighted, added or changed text in the main manuscript or SI in green and *italic*. Original text from the manuscript is shown in *italic*.

I note that a study similar to the meta-analysis referred to in ref. 33 is now available in the same journal, but using a much vaster database, which provides more solid evidence for the non-triviality of the existence of a correlation between activation and binding free energies. As a final note, I also suggest the authors include a brief paragraph in the final manuscript with the answer for question g2, as I believe this question will come to the mind to many readers.

We thank the reviewer for the positive feedback. We have now included the suggested reference and added the following text in the discussion (main text, page 9).

The existence of LFERs for enzyme reactions governed by the chemical step remains to be investigated further, but meta-analyses of kinetic databases show little correlation between k_{cat} and K_M (33,35,50). This is unlike many reactions in both homogenous and heterogeneous (non-biochemical) catalysis which may be limited by a LFER even though the reaction is governed by a chemical step (51,52). Kinetic parameters for heterogeneous enzyme reactions are scarce. Thus, it is still an open question whether scaling relations are as common in heterogeneous biocatalysis as it is in inorganic heterogeneous catalysis (54) but the current study show that cellulases are severely restricted by a LFER.

References

1. Medford, A. J., Vojvodic, A., Hummelshøj, J. S., Voss, J., Abild-Pedersen, F., Studt, F., Bligaard, T., Nilsson, A., and Nørskov, J. K. (2015) From the Sabatier principle to a predictive theory of transition-metal heterogeneous catalysis. *Journal of Catalysis* **328**, 36-42
2. Kari, J., Christensen, S. J., Andersen, M., Baiget, S. S., Borch, K., and Westh, P. (2019) A practical approach to steady-state kinetic analysis of cellulases acting on their natural insoluble substrate. *Anal Biochem* **586**, 113411
3. Cruys-Bagger, N., Alasepp, K., Andersen, M., Ottesen, J., Borch, K., and Westh, P. (2016) Rate of Threading a Cellulose Chain into the Binding Tunnel of a Cellulase. *Journal of Physical Chemistry B* **120**, 5591-5600
4. Segel, I. H. (1975) *Enzyme kinetics: Behavior and analysis of rapid equilibrium and steady-state enzyme systems*, Wiley, New York
5. Cruys-Bagger, N., Ren, G., Tatsumi, H., Baumann, M. J., Spodsberg, N., Andersen, H. D., Gorton, L., Borch, K., and Westh, P. (2012) An amperometric enzyme biosensor for real-time measurements of cellobiohydrolase activity on insoluble cellulose. *Biotechnology and bioengineering* **109**, 3199-3204
6. Kipper, K., Väljamäe, P., and Johansson, G. (2005) Processive action of cellobiohydrolase Cel7A from *Trichoderma reesei* is revealed as 'burst' kinetics on fluorescent polymeric model substrates. *Biochem. J.* **385**, 527-535
7. Praestgaard, E., Elmerdahl, J., Murphy, L., Nymand, S., McFarland, K. C., Borch, K., and Westh, P. (2011) A kinetic model for the burst phase of processive cellulases. *The FEBS journal* **278**, 1547-1560
8. Jalak, J., and Valjamae, P. (2010) Mechanism of initial rapid rate retardation in cellobiohydrolase catalyzed cellulose hydrolysis. *Biotechnology and bioengineering* **106**, 871-883
9. Kurasin, M., and Valjamae, P. (2011) Processivity of cellobiohydrolases is limited by the substrate. *The Journal of biological chemistry* **286**, 169-177
10. Cruys-Bagger, N., Elmerdahl, J., Praestgaard, E., Tatsumi, H., Spodsberg, N., Borch, K., and Westh, P. (2012) Pre-steady state kinetics for the hydrolysis of insoluble cellulose by cellobiohydrolase Cel7A. *The Journal of biological chemistry* **287**
11. Cruys-Bagger, N., Tatsumi, H., Ren, G. R., Borch, K., and Westh, P. (2013) Transient kinetics and rate-limiting steps for the processive cellobiohydrolase Cel7A: effects of substrate structure and carbohydrate binding domain. *Biochemistry* **52**, 8938-8948
12. Murphy, L., Cruys-Bagger, N., Damgaard, H. D., Baumann, M. J., Olsen, S. N., Borch, K., Lassen, S. F., Sweeney, M., Tatsumi, H., and Westh, P. (2012) Origin of initial burst in activity for *Trichoderma reesei* endo-glucanases hydrolyzing insoluble cellulose. *The Journal of biological chemistry* **287**, 1252-1260
13. Christensen, S. J., Kari, J., Badino, S. F., Borch, K., and Westh, P. (2018) Rate-limiting step and substrate accessibility of cellobiohydrolase Cel6A from *Trichoderma reesei*. *The FEBS journal* **285**, 4482-4493
14. Cruys-Bagger, N., Elmerdahl, J., Praestgaard, E., Borch, K., and Westh, P. (2013) A steady-state theory for processive cellulases. *The FEBS journal* **280**, 3952-3961
15. Kari, J., Olsen, J., Borch, K., Cruys-Bagger, N., Jensen, K., and Westh, P. (2014) Kinetics of Cellobiohydrolase (Cel7A) Variants with Lowered Substrate Affinity. *The Journal of biological chemistry* **289**, 32459-32468
16. Røjel, N., Kari, J., Sørensen, T. H., Badino, S. F., Morth, J. P., Schaller, K., Cavaleiro, A. M., Borch, K., and Westh, P. (2019) Substrate binding in the processive cellulase Cel7A: Transition state of complexation and roles of conserved tryptophan residues.

17. Cruys-Bagger, N., Elmerdahl, J., Praestgaard, E., Tatsumi, H., Spodsberg, N., Borch, K., and Westh, P. (2012) Pre-steady-state kinetics for hydrolysis of insoluble cellulose by cellobiohydrolase Cel7A. *The Journal of biological chemistry* **287**, 18451-18458
18. Warshel, A. (1998) Electrostatic Origin of the Catalytic Power of Enzymes and the Role of Preorganized Active Sites. *J. Biol. Chem.* **273**, 27035-27038
19. Fersht, A. R., and Sato, S. (2004) Phi-value analysis and the nature of protein-folding transition states. *Proc Natl Acad Sci U S A* **101**, 7976-7981
20. Kari, J., Schiano-di-Cola, C., Hansen, S. F., Badino, S. F., Sørensen, T. H., Cavaleiro, A. M., Borch, K., and Westh, P. (2020) A steady-state approach for inhibition of heterogeneous enzyme reactions. *Biochem J* **477**, 1971-1982

REVIEWER COMMENTS

Reviewer #1 (Remarks to the Author):

I was asked by the editor to read the response from the authors to Reviewer 2, who brought up many excellent points and wrote an exceptionally detailed and thoughtful set of reviews.

Overall, I still maintain that this is one of the best cellulase papers, and interfacial biocatalysis papers, I have ever read. I am excited to see it published as soon as possible, so I strongly recommend that the paper be accepted immediately.

Reviewer #2 (Remarks to the Author):

I hope my review is clear. It is uploaded as a PDF file which I hope can be read clearly. If you have questions, just let me know. This has been a lengthy procedure. On my part, this current review is final, at least for manuscripts reasonably similar to the original version (and the two revisions that have resulted). Up to know, the authors have responded to my concerns, but without convincing me, and the manuscript has remained largely what is was originally.

(Please see file on the next page)

The authors have provided detailed rebuttal to my review. Besides specific responses (see below), they reply with a general comment whose essence appears to be this: for the claims of the manuscript to be valid, the kinetic model/mechanism of the enzymatic reaction is not relevant. They thus use the phenomenological interpretation of the K_m (i.e., K_m is the substrate concentration at which the rate is half of the maximum) for a common basis. They suggest this to reconcile controversial points.

I cannot agree with the authors. Interpretation of the data is within the authors' discretion, but the resulting claims must be clear and adequately supported. I have commented in two previous reviews that the manuscript raises concern in these respects. The reviews have resulted in a lengthy debate but regrettably, in only minimal changes of the manuscript. My original impression, that the manuscript involves overinterpretation of a relatively simple set of kinetic data, has not changed.

1) I have no qualms about the use of the phenomenological interpretation of the K_m . But the manuscript has to be consistent. The authors know from literature (including their own earlier work), and Figure also shows it, that this K_m involves "kinetic complexity" to a considerable extent. Therefore, it simply does not make sense to calculate free energies of binding, or to draw kinetic barrier diagrams for the reaction coordinate, based on K_m values as half-saturation constants. The correlation between the maximum rate (k_{cat}) and the K_m can be shown as observed, preferably in linear form (see later). But the appeal to free energies of binding based on half-saturation constants is misleading. The observation of burst kinetics in all of these enzymes (see the rebuttal on page 4) should serve as important note of caution.

2) The authors seem to suggest that their claims follow directly from the observations, based on an interpretation that is both clear (i.e., unambiguous) and independent of the assumed kinetic mechanism. However, various scenarios of fundamentally different mechanistic origin can lead to a linear correlation between the K_m and the k_{cat} . One important possibility is nonproductive binding. Consider enzymes bound on the solid cellulose at different "sites", of which only some/few enable catalysis. Only to note, the cellulose surface of the substrate particles will certainly not be uniform in many of its structural parameters. The effect of nonproductive binding is that the k_{cat} and the K_m are both decreased by the enzyme portion that has been bound nonproductively. With nonproductive binding as an important factor, one could easily imagine the observed trend for k_{cat} and K_m to arise for different enzymes (wildtypes, variants) irrespective of their structure and mechanism. To be clear, it is not my point to adjudicate the plausibility of the different interpretations. My point is only to direct attention to *their plausible existence*. This in itself should be enough to re-assess a single (preferred) line of interpretation. It won't escape attention that all further claims (see Fig. 5) are completely relative to the main interpretation of the data. If nonproductive binding is a relevant factor, the discussion about ground-state stabilization as opposed to transition stabilization has no longer significance. Physical constraints are not evident. If relevant at all, discussion about enzyme evolution would have to take a different course. Conclusion at this point must therefore be: in the absence of a well-supported kinetic-mechanistic model of the enzymatic process, the trend of the k_{cat} and K_m data cannot be interpreted with reasonable clarity.

3) There seems to be consensus in the literature that the k_{cat} of the GH7 cellobiohydrolase from *Trichoderma reesei* is limited by the release of the enzyme post-catalysis. The exact

rate at which this dissociation happens, however, is less clear. Following the justifiable argument of the authors in the rebuttal, that the substrate is no longer the same after the catalytic reaction, I call the dissociation-ready state of the enzyme the “enzyme-product complex”. If they prefer, one can call it the “modified enzyme-substrate complex”. I have rejected in my last review the idea that enzyme dissociation from substrate and product (modified substrate) can a priori be assigned the same rate constant. Evidence that catalytic enzyme variants show a similar desorption rate as the wildtype only reveals that dissociation of the variant from the enzyme-substrate occurs at a comparable rate as dissociation of the wildtype enzyme from the product complex. The wildtype enzyme may show fluorescence in both substrate and product complexes or the fluorescence is from the small portion of enzyme in the substrate complex. And even if both rate constants of dissociation are of similar magnitude (as the authors consistently argue), the rate-controlling dissociation in the enzymes happens from the “product state” because catalysis is fast. This brings us back to the issue of K_m and substrate binding energy.

4) I agree with the authors’ notion that in general, the experimental K_m and k_{cat} could be scattered anywhere in the k_{cat} vs. K_m plot. However, the fact, that the points lie approximately along a line (or a curve) suggests that there is only one dominant underlying parameter that changes among the enzymes. From what was stated above (#2), the fraction of productively bound enzyme on the cellulose surface could be one such parameter. Assuming the kinetic model of Scheme S1 and considering literature on some of these enzymes, one is directed to the dissociation rate of the “enzyme-product complex”. Evidences on the GH7 cellobiohydrolases, including a number of papers from the group of the authors, support the idea that change in the k_{cat} would arise from change in the rate-limiting dissociation rate. Under these circumstances, any change in k_{cat} thus caused implies a proportional change in the K_m , based on relationships deducible from the kinetic mechanism. The authors now argue that the K_m would not have to change as it did because other rate constants could have also changed. I agree completely, but one is reminded at the same time on the principle of parsimony. Why would one assume more changes when a single change already explained the observed effect entirely? At least for the GH7 cellobiohydrolases which constitute a large group of the cellulases analyzed in their selection of enzymes, the immediate relationship between k_{cat} and K_m is strongly implied from their earlier papers (see my previous review).

It is my impression that except for insisting on the same dissociation rate constant for release of enzyme from substrate and product complex, the position of the authors is not fundamentally different from mine as regards the use of Scheme S1. We seem to agree on the notion that slow release of enzyme could be a reason (note: not the only one!) for the observed trend in k_{cat} and K_m . I understand their reluctance to assign a separate rate constant to each dissociation step (see my previous review), for if one did, one would also have to abandon the idea of an LFER based on K_m as parameter to calculate binding energies. This would also make Figure 5 and the associated discussion unnecessary. The mysterious discussion of why an enzyme moves up and down the “line” in the plot of k_{cat} and K_m could be replaced by clarity if the concept of rate-limiting product dissociation was consistently applied.

I have now reviewed the manuscript three times. In my assessment, the predicament is inescapable. The discussion of the manuscript has to be revised conceptually at its core.

5) I appreciate their efforts to determine with a real-time method the kinetics of the GH45 endoglucanase. But the numbers seem not clear to me. The integrated “burst” in Figure S3 corresponds to more than 25 turnovers of the available enzyme. Therefore, it can hardly be explained by a rate-limiting step (before or after the product release) in a single hydrolysis cycle. The calculated k_{cat} (= volumetric rate/enzyme molarity) is 0.11 per sec (= $0.011 \mu\text{M s}^{-1}/0.1 \mu\text{M enzyme}$) or 6.6 per min. Thus, even at its steady state rate, the enzyme undergoes several turnovers in the time assigned to the burst. I don’t see why the enzyme couldn’t have been desorbed and re-adsorbed in the burst phase. From the results shown, it is not clear what causes the decline in the rate. The fact that cellulose is not a uniform substrate can be relevant. They also mention in text the accumulation of unproductively bound enzymes on the surface. Please refer to what was said above for the effect of nonproductive binding on k_{cat} and K_m .

6) Lastly, I would like to comment on the implied practical application. I have mentioned in my first review that I was not convinced. Due to the debate about the mechanistic interpretation of the data, this aspect has faded into the background. I have taken the results from Table S1 and plotted them in a linear fashion, as shown below. For all practical purposes, this is the relevant “scaling relationship”, not the double-log plot used throughout the manuscript. I have eliminated from the plot the outliers marked by the authors in Table S1 and additionally removed the three K_m values at over 300 g/L. K_m values as high as these can hardly be relevant and they tend to influence the correlation negatively. From the plot below one notices a trend, but one with very considerable scatter. Prediction of the k_{cat} from the K_m may be possible, but only with substantial degree of uncertainty. I can be debated whether a correlation like this supports an in-silico screening program for enzymes with tunable kinetic parameters.

Figure: Lin-linear plot of the kinetic data from Table S1, with marked outliers and K_m values ≥ 300 g/L excluded. The line is from linear regression.

It is well known that in order to assess enzymes for practical application one should examine the (specific) activity at the technologically relevant substrate concentration (e.g., the substrate at solubility limit, ...). Of a set of enzymes having the same/similar specificity constant (k_{cat}/K_m), one can select the one that gives the highest rate. If the relationship

between k_{cat} and K_m is well established, one can also use the mathematical transformation on the Michaelis-Menten equation, as shown by the authors, to calculate the “best K_m enzyme” for fastest initial rate under the conditions used. But all these considerations are extremely biased towards only one side of the application (i.e., the initial rate). It is well known that enzymes with relatively high K_m have severe problems in achieving high substrate conversions efficiently. This is trivially explained from the hyperbolic relationship of rate vs. substrate concentration or the corresponding integrated relationship. One could easily calculate the “best K_m enzyme” for achieving the desired conversion of the cellulose substrate (at defined initial concentration) in the shortest possible time. For a real application (note: to hydrolyze Avicel model substrate!), this information would be much more valuable than Figure 6. From a practical point of view, productivity (at time to a desired conversion/product concentration) is primary to initial reaction rate. I am thus very skeptic about their optimum K_m as 2.8 times the initial substrate concentration. Selecting such kind of enzyme, it would be impossible to hydrolyze a substantial portion of the substrate (say, $\geq 80\%$) efficiently. In general, however, any such simple calculation, whether it is rate or productivity, should be subjected to experimental verification.

Response to reviewer 2

Again we would like to thank the reviewer for the thorough feedback. Although this has been an exhausting revision we are grateful for the scientific discussion that has originated from the persistent and committed revision of the reviewer. The core of the dispute still concern the meaning/interpretation of the reported apparent Michaelis-Menten constants (K_M) and the subsequent discussion of the origin of the correlation between the apparent MM-parameters (K_M and k_{cat}). We agree with the reviewer that one must be cautious when interpreting apparent K_M for a composite enzyme reaction and that our approach is an interpretation of the main observable which is the scaling between the apparent K_M and k_{cat} . We have now made substantial changes in the main manuscript (and figures) to acknowledge the main concerns raised by the reviewer. Specifically, we have reorganized the manuscript such that the observations (e.g correlation between K_M and k_{cat}) are presented in the results section and the interpretation (LFER) have been moved to the discussion. Additionally, we have emphasized the potential pitfalls of using the apparent K_M as an affinity descriptor and reference our effort to justify our interpretation.

Below is a point-by-point answer to the questions and concerns raised by the reviewer. The original comments from the referees are shown in **red**, while our reply is shown without any change in font or color. Changed in the text are shown in **green** and deletions are shown with ~~strikethrough~~. Unchange text is shown in *italic* font.

Specific questions

In the following, we will reply to each of the questions and concerns.

- 1) I have no qualms about the use of the phenomenological interpretation of the K_M . But the manuscript has to be consistent. The authors know from literature (including their own earlier work), and Figure also shows it, that this K_M involves “kinetic complexity” to a considerable extent. Therefore, it simply does not make sense to calculate free energies of binding, or to draw kinetic barrier diagrams for the reaction coordinate, based on K_M values as half-saturation constants. The correlation between the maximum rate (k_{cat}) and the K_M can be shown as observed, preferably in linear form (see later). But the appeal to free energies of binding based on half-saturation constants is misleading. The observation of burst kinetics in all of these enzymes (see the rebuttal on page 4) should serve as important note of caution.

This point raised by the reviewer has been the core of the discussion since the first review. We have taken this critique very seriously and in both revisions made additional experiments, simulations, and presented theoretical arguments rooted in the current knowledge about cellulase kinetics. We are convinced that this additional work supported our interpretations of the K_M even though we agree with the reviewer that one must be careful when interpreting an apparent K_M for a composite enzyme reaction. We agree with the reviewer that our approach is a (supported) interpretation and for this reason, we have now made significant changes in the main manuscript. Specifically, we have reorganized the manuscript such that the Results section present the correlation between K_M and k_{cat} as it is directly observed in the experiments, while the energetic interpretation of the parameters has been move to the discussion.

In the section “Kinetic analysis” (page 3) we present the correlation between Km and kcat without any interpretation of the parameters. Specifically, we changed

~~The relationship between steady-state parameters from a more realistic model and those from the simple MM framework is further discussed in the SI. The derived rates were based on soluble products only and control experiments (Table S3, SI) showed that this was a good descriptor of the overall activity even for EGs. We used k_{cat} and K_M to estimate changes in respectively transition-state free energy ($-\Delta\Delta G^\ddagger$) and standard free energy of ligand binding ($-\Delta\Delta G_B^o$) (c.f. Fig. 5) following well-established principles (1-3). Advantages and limitation of K_M as a descriptor of the enzyme-substrate affinity are further discussed in the SI (see figure S6, SI). Specifically, we used the equations~~

~~$$-\Delta\Delta G_B^o = RT \ln \left(\frac{K_M}{K_{M,ref}} \right) \quad (\text{Eq. 2})$$~~

~~$$-\Delta\Delta G^\ddagger = -RT \ln \left(\frac{k_{cat}}{k_{cat,ref}} \right) \quad (\text{Eq. 3})$$~~

~~Equations 2 and 3 introduce a reference enzyme with the kinetic parameters $K_{M,ref}$ and $k_{cat,ref}$, and hence the calculated free energies are energy changes relative to the selected reference. This approach alleviates ambiguities regarding standard states (Eq. 2) and pre-exponential factors (Eq. 3). We used the GH6 cellobiohydrolase from *Trichoderma reesei* (TrCel6A) as our reference enzyme and it follows that this enzyme will have (i.e. TrCel6A will be located in the origin of Fig. 2 below). We emphasize that the selection of reference enzyme merely serves to define an origin; it has no bearing on the subsequent discussion, which considers changes in the free energy.~~

~~In Fig. 2, the change in activation free energies ($-\Delta\Delta G^\ddagger$) the natural logarithm of the derived kinetic parameters (K_M and k_{cat}) are plotted against each other the change in standard binding free energies ($-\Delta\Delta G_B^o$) for all investigated enzymes to illustrate the power-law correlation between the kinetic parameters.~~

The main figure (Fig. 2) was change to show the correlation between Km and kcat.

Fig. 2. Correlation plot of $\ln(K_M)$ and $\ln(k_{cat})$. *change in the binding free energy ($\Delta\Delta G_B^\circ$, Eq. 2) and activation free energy ($-\Delta\Delta G^\ddagger$, Eq. 3) for all...*

The section “computational analysis” (page 5) has also been rewritten such that the computational analysis now serve as a computer experiment to test the hypothesis that K_M may be interpreted as an affinity parameter.

The section has been change from:

The strong correlation between binding- and activation- free energies shown in Fig. 2, may open up for prediction of catalytic rates based solely on computed binding free energies. To test this approach, we calculated cellulose binding strengths for a subset of nine enzymes from Fig. 2, using molecular dynamics (MD) simulations with umbrella sampling along the binding path (See further details in figure S8, SI). We selected enzymes that spanned a wide range of $\Delta\Delta G_B^\circ$ and represented all structural and functional classes listed in Fig. 1. For modular cellulases, the contribution of the CBM to $\Delta\Delta G_B^\circ$ was computed separately. Comparisons in Fig. 3 showed that despite the diversity of the analyzed cellulases, computed changes in binding affinity, $\Delta\Delta G_{B,MD}^\circ$, scaled reasonably with the experimental values, $\Delta\Delta G_{B,Exp}^\circ$.

Fig. 3. Correlation of computed ($\Delta\Delta G_{B, MD}$) and experimental ($\Delta\Delta G_B^\circ$) changes in free energies of binding for nine selected cellulases. Experimental values are taken from Fig. 2. The selected cellulases covered a wide range of $\Delta\Delta G_B^\circ$ shown in Fig. 2 and encompassed all main structural and functional traits specified in Fig. 1.

To the following:

Computational analysis.

The strong correlation between $\ln(K_M)$ and $\ln(k_{cat})$ shown in Fig. 2 is attractive from a computational point of view. If the apparent Michaelis-Menten constant (K_M) can be interpreted as a descriptor for the enzyme-substrate binding affinity it may open up for prediction of catalytic rates based solely on computed binding free energies. To test this hypothesis we computed cellulose binding strengths for a subset of nine enzymes from Fig. 2, using molecular dynamics (MD) simulations with umbrella sampling along the binding path (See further details in Fig. S8, SI). We selected enzymes that spanned a wide range of K_M values and represented all structural and functional classes listed in Fig. 1. For modular cellulases, the contribution of the CBM to binding energy ΔG_B° was computed separately. To compare with experiments we used k_{cat} and K_M , to estimate changes in respectively transition-state free energy ($\Delta\Delta G^\ddagger$) and standard free energy of ligand binding ($\Delta\Delta G_B^\circ$) following well-established principles (1-3). Specifically, we used the equations

$$\Delta\Delta G_B^\circ = RT \ln \left(\frac{K_M}{K_{M,ref}} \right) \quad (\text{Eq. 2})$$

$$\Delta\Delta G^\ddagger = -RT \ln \left(\frac{k_{cat}}{k_{cat,ref}} \right) \quad (\text{Eq. 3})$$

which introduce a reference enzyme with the kinetic parameters $K_{M,ref}$ and $k_{cat,ref}$. Hence, the calculated free energies are energy changes relative to the selected reference. This approach alleviates ambiguities regarding standard states (Eq. 2) and pre-exponential factors (Eq. 3). We used the GH6 cellobiohydrolase from *Trichoderma reesei* (TrCel6A) as our reference enzyme and it follows that this enzyme will have $\Delta\Delta G^\ddagger = \Delta\Delta G^\circ_B = 0$.

The validity of equation 2 is dependent on whether K_M can be interpreted as a descriptor of the enzyme-substrate affinity. The comparison in Fig. 3a showed that despite the diversity of the analyzed cellulases, computed changes in binding affinity, $\Delta\Delta G_{B,MD}$, scaled reasonably well with the experimental values, $\Delta\Delta G_{B,exp}$, derived from equation 2. This supports the validity of eq. 2 and the idea of using computed ligand-binding energies to predict catalytic rates (e.g. $\Delta\Delta G_{B,MD} \rightarrow \Delta\Delta G_{B,exp} \rightarrow \Delta\Delta G^\ddagger_{exp}$). Fig. 3b illustrate the scaling between $\Delta\Delta G_{B,MD}$ and $\Delta\Delta G^\ddagger_{exp}$.

Fig. 3. Correlation of computed and experimental free energies for nine selected cellulases. a) Changes in computed free energies of binding ($\Delta\Delta G_{B,MD}$) and experimental changes in binding free energy ($\Delta\Delta G_{B,exp}$). b) Correlation $\Delta\Delta G_{B,MD}$ and experimental changes in activation free energy ($\Delta\Delta G^\ddagger_{exp}$). Experimental free energies were calculated using eq. 2 and 3. The kinetic parameters (K_M and k_{cat}) of the nine cellulases can be found in in Tab. S4 (SI). The selected cellulases covered a wide range of kinetic parameters shown in Fig. 2 and encompassed all main structural and functional traits specified in Fig. 1. Standard deviations of the experimental free energies ($\Delta\Delta G_{B,exp}$ and $\Delta\Delta G^\ddagger_{exp}$) and computed free energies are shown as error bars.

- 2) The authors seem to suggest that their claims follow directly from the observations, based on an interpretation that is both clear (i.e., unambiguous) and independent of the assumed kinetic mechanism. However, various scenarios of fundamentally different mechanistic origin can lead to a linear correlation between the K_M and the k_{cat} . One important possibility is nonproductive binding. Consider enzymes bound on the solid cellulose at different “sites”, of which only some/few enable catalysis. Only to note, the cellulose surface of the substrate particles will certainly not be uniform in many of its structural parameters. The effect of nonproductive binding is that the k_{cat} and the K_M are both decreased by the enzyme portion that has been bound

nonproductively. With nonproductive binding as an important factor, one could easily imagine the observed trend for k_{cat} and K_M to arise for different enzymes (wildtypes, variants) irrespective of their structure and mechanism. To be clear, it is not my point to adjudicate the plausibility of the different interpretations. My point is only to direct attention to their plausible existence. This in itself should be enough to re-assess a single (preferred) line of interpretation. It won't escape attention that all further claims (see Fig. 5) are completely relative to the main interpretation of the data. If nonproductive binding is a relevant factor, the discussion about ground-state stabilization as opposed to transition stabilization has no longer significance. Physical constraints are not evident. If relevant at all, discussion about enzyme evolution would have to take a different course. Conclusion at this point must therefore be: in the absence of a well-supported kinetic-mechanistic model of the enzymatic process, the trend of the k_{cat} and K_M data cannot be interpreted with reasonable clarity.

As explained in the following two answers (3 and 4) we do not see how our interpretation is fundamentally different from the one presented by the reviewer. We agree with the reviewer that the molecular origin of the scaling is not well understood but our (supported) interpretation is one way to rationalize the results. However, the particular origin of the scaling between k_{cat} and K_M is needed neither for the discussion of the restricted fitness landscape for cellulases nor for considerations of practical use e.g. in enzyme design. To acknowledge the criticism by the reviewer and generalize the discussion we have now changed the first part of the discussion ("enzyme fitness and physical constraints") so that it is based on the observed scaling between K_M and k_{cat} .

Specifically we change the first paragraph in the discussion (page 7) and fig. 4.

Enzyme fitness and physical constraints. Fig. 2 ~~presents~~ may be seen as a fitness landscape for cellulases attacking their native insoluble substrate, and it appears that most enzymes accumulated around the diagonal. The diagonal defines a continuum ranging from enzymes with weak substrate interactions and rapid turnover (high K_M and k_{cat}), to enzymes with stronger interactions, but slower turnover (low K_M and k_{cat}). This defines a continuum of properties ranging from enzymes with high substrate affinity but a low turnover rate (upper left end, Fig. 2A), to enzymes with a low substrate affinity but a high turnover number (lower right end, Fig. 2A).

Fig. 4. Illustration of the effect of the non-catalytic CBM (panel a) and tryptophan residues in the catalytic domain of cellobiohydrolases from GH6 and GH7 (panel b). Correlation plot of $\ln(K_M)$ and $\ln(k_{cat})$ for ...

- 3) There seems to be consensus in the literature that the k_{cat} of the GH7 cellobiohydrolase from *Trichoderma reesei* is limited by the release of the enzyme post-catalysis. The exact rate at which this dissociation happens, however, is less clear. Following the justifiable argument of the authors in the rebuttal, that the substrate is no longer the same after the catalytic reaction, I call the dissociation-ready state of the enzyme the “enzyme-product complex”. If they prefer, one can call it the “modified enzyme-substrate complex”. I have rejected in my last review the idea that enzyme dissociation from substrate and product (modified substrate) can a priori be assigned the same rate constant. Evidence that catalytic enzyme variants show a similar desorption rate as the wildtype only reveals that dissociation of the variant from the enzyme-substrate occurs at a comparable rate as dissociation of the wildtype enzyme from the product complex. The wildtype enzyme may show fluorescence in both substrate and product complexes or the fluorescence is from the small portion of enzyme in the substrate complex. And even if both rate constants of dissociation are of similar magnitude (as the authors consistently argue), the rate-controlling dissociation in the enzymes happens from the “product state” because catalysis is fast. This brings us back to the issue of K_M and substrate binding energy.

We do not see any conflict here. We have consistently argue that the rate-controlling dissociation happens from the post-catalysis enzyme complex, which the reviewer calls enzyme-product complex. If we view the reactions as an *ordered uni-bi reaction* (scheme S1 in the response letter for the second revision) the steady-state solution give a K_M that is identical to the dissociation constant ($K_M = k_{off}/k_{on}$) assuming that;

- dissociation prior and post catalysis is govern by the same rate-constant (supported by measurements with inactive enzymes).
- hydrolysis (actual catalysis) is much faster than the dissociation (supported by literature)

We refer to reponse letter 2.0 (eq. S3) for further details.

- 4) I agree with the authors' notion that in general, the experimental K_m and k_{cat} could be scattered anywhere in the k_{cat} vs. K_m plot. However, the fact, that the points lie approximately along a line (or a curve) suggests that there is only one dominant underlying parameter that changes among the enzymes. From what was stated above (#2), the fraction of productively bound enzyme on the cellulose surface could be one such parameter. Assuming the kinetic model of Scheme S1 and considering literature on some of these enzymes, one is directed to the dissociation rate of the "enzyme-product complex". Evidences on the GH7 cellobiohydrolases, including a number of papers from the group of the authors, support the idea that change in the k_{cat} would arise from change in the rate-limiting dissociation rate. Under these circumstances, any change in k_{cat} thus caused implies a proportional change in the K_m , based on relationships deducible from the kinetic mechanism. The authors now argue that the K_m would not have to change as it did because other rate constants could have also changed. I agree completely, but one is reminded at the same time on the principle of parsimony. Why would one assume more changes when a single change already explained the observed effect entirely? At least for the GH7 cellobiohydrolases which constitute a large group of the cellulases analyzed in their selection of enzymes, the immediate relationship between k_{cat} and K_m is strongly implied from their earlier papers (see my previous review).

It is my impression that except for insisting on the same dissociation rate constant for release of enzyme from substrate and product complex, the position of the authors is not fundamentally different from mine as regards the use of Scheme S1. We seem to agree on the notion that slow release of enzyme could be a reason (note: not the only one!) for the observed trend in k_{cat} and K_m . I understand their reluctance to assign a separate rate constant to each dissociation step (see my previous review), for if one did, one would also have to abandon the idea of an LFER based on K_m as parameter to calculate binding energies. This would also make Figure 5 and the associated discussion unnecessary. The mysterious discussion of why an enzyme moves up and down the "line" in the plot of k_{cat} and K_m could be replaced by clarity if the concept of rate-limiting product dissociation was consistently applied.

I have now reviewed the manuscript three times. In my assessment, the predicament is inescapable. The discussion of the manuscript has to be revised conceptually at its core.

We agree. Our main interpretations rely on the same idea and is not in conflict with the explanation given be the reviewer. The underlying assumption is that surface obstacles hinder the processive hydrolysis such that most of the enzymes (at steady-state) are in an unproductive (post-catalysis) ES complex (referred to as ES^* in the previous response letter). It is important to note that ES and ES^* is similar for cellulases since cellulases have between 5-9 pyranose subsites (see figure below).

This is very different from most enzymes where the post-catalysis complex only include the product (EP complex). Binding isotherms and binding kinetics with inactive variants support the idea that dissociation from the ES and ES^* state can be assigned to the same rate-constant. This indeed simplify the steady-state kinetics and as explained in the response letter for the second revision (scheme S2 and eq. S3 in response letter 2) this may allow an interpretation of K_m as a dissociation constant for the stability of all ES complex (ES and ES^*). However, if separate rate-constant are inferred for the dissociation of ES and

ES* the K_M may be viewed as a descriptor for the stability of the ES* complex (eq. S5 in response letter 2). In both cases the rate-limiting step seems to be governed by a slow dissociation rate-constant for the unproductive ES* complex, which means that the rate-constant for the dissociation from ES* will appear in both K_M and V_{max}/E_0 . This is our interpretation of the observed scaling between K_M and k_{cat} . This interpretation can also be translated into an energy representation of the same phenomenon where the energy barrier associated with this rate-constant is linked to the depth of the energy well for the ES* complex (e.g. $dG_{binding} = dG_{++}(k_{off}) - dG_{++}(k_{on})$).

We agree with the reviewer that one must be careful when interpreting an apparent K_M for a composite enzyme reaction and for that reason we have now added the following text in the section "Origin of physical constraint" in the discussion to emphasize this point.

Specifically we have now added this text after eq. 4 (page 8):

where Φ is a scaling constant that converts changes in the binding free energy ($\Delta\Delta G_B$) to changes in activation free energy ($\Delta\Delta G^\ddagger$). The correlation shown in Fig. 2 may be interpreted as an LFER if the K_M values can be interpreted as a dissociation constant for the enzyme-substrate complex. In general one has to be cautious when using the (apparent) K_M value as an affinity descriptor for complex enzyme reactions such as the one studied here. However, such interpretation of K_M has been successfully used earlier (1-3) and it is also in line with the MD results (Fig. 3a) that showed good correlations between computed ligand-binding energies and experimental binding energies calculated using Eq. 2. The validity of K_M as a descriptor of the enzyme-substrate affinity of the investigated enzymes is further discussed in the SI (see Fig. S6, SI).

Using Eq. 2 and 3 we calculated $\Delta\Delta G_B^\circ$ and $\Delta\Delta G^\ddagger$ and found that the two free energies correlated with a slope of $\Phi = -0.74 \pm 0.02$ (see SI, Fig. S9). This is the same slope as found for the line in Fig. 2 but with opposite sign due to the minus in eq. 3 (e.g. high k_{cat} values give low activation energies).

- 5) I appreciate their efforts to determine with a real-time method the kinetics of the GH45 endoglucanase. But the numbers seem not clear to me. The integrated "burst" in Figure S3 corresponds to more than 25 turnovers of the available enzyme. Therefore, it can hardly be explained by a rate-limiting step (before or after the product release) in a single hydrolysis cycle. The calculated k_{cat} (= volumetric rate/enzyme molarity) is 0.11 per sec (= $0.011 \mu\text{M s}^{-1}/0.1 \mu\text{M}$ enzyme) or 6.6 per min. Thus, even at its steady state rate, the enzyme undergoes several turnovers in the time assigned to the burst. I don't see why the enzyme couldn't have been desorbed and re-adsorbed in the burst phase. From the results shown, it is not clear what causes the decline in the rate. The fact that cellulose is not a uniform substrate can be relevant. They also mention in text the accumulation of unproductively bound enzymes on the surface. Please refer to what was said above for the effect of nonproductive binding on k_{cat} and K_M .

Little is known about the kinetics of GH45 but many endoglucanases are endo-processive enzymes that can do several consecutive hydrolytic events in a single processive hydrolysis cycle. For this reason the bursts are expected to be much higher than a conventional ordered uni-bi reaction with one product released per enzyme molecule per cycle. We agree that further studies are needed to elucidate the origin

of this phenomenon, but it serves to show that the rate-limiting step probably lies after the hydrolytic step.

- 6) Lastly, I would like to comment on the implied practical application. I have mentioned in my first review that I was not convinced. Due to the debate about the mechanistic interpretation of the data, this aspect has faded into the background. I have taken the results from Table S1 and plotted them in a linear fashion, as shown below. For all practical purposes, this is the relevant scaling relationship, not the double-log plot used throughout the manuscript. I have eliminated from the plot the outliers marked by the authors in Table S1 and additionally removed the three K_m values at over 300 g/L. K_m values as high as these can hardly be relevant and they tend to influence the correlation negatively. From the plot below one notices a trend, but one with very considerable scatter. Prediction of the k_{cat} from the K_m may be possible, but only with substantial degree of uncertainty. It can be debated whether a correlation like this supports an in-silico screening program for enzymes with tunable kinetic parameters.

Figure: Lin-linear plot of the kinetic data from Table S1, with marked outliers and K_m values ≥ 300 g/L excluded. The line is from linear regression.

It is well known that in order to assess enzymes for practical application one should examine the (specific) activity at the technologically relevant substrate concentration (e.g., the substrate at solubility limit,). Of a set of enzymes having the same/similar specificity constant (k_{cat}/K_m), one can select the one that gives the highest rate. If the relationship between k_{cat} and K_m is well established, one can also use the mathematical transformation on the Michaelis-Menten equation, as shown by the authors, to calculate the “best K_m enzyme” for fastest initial rate under the conditions used. But all these considerations are extremely biased towards only one side of the application (i.e., the initial rate). It is well known that enzymes with relatively high K_m have severe problems in achieving high substrate conversions efficiently. This is trivially explained from the hyperbolic relationship of rate vs. substrate concentration or the corresponding integrated relationship. One could easily calculate the “best K_m enzyme” for achieving the desired conversion of the cellulose substrate (at defined initial concentration) in the shortest possible time. For a real application (note: to hydrolyze Avicel model substrate!), this information would be much more valuable than Figure 6. From a practical point of view, productivity (at time to a desired conversion/product concentration) is primary to initial reaction rate. I am thus very skeptic about their optimum K_m as 2.8 times the initial substrate concentration. Selecting such kind of enzyme, it would be impossible to hydrolyze a substantial portion of the substrate (say, \geq

80%) efficiently. In general, however, any such simple calculation, whether it is rate or productivity, should be subjected to experimental verification.

It is true that the correlation between K_m and k_{cat} is close to linear in a normal plot that is why we find a slope in the log-log plot close to 1. However, a power law fit with two parameters fits the K_m vs. k_{cat} data better than a simple line with two parameter. This is illustrated in the figure below where the R^2 for the power law fit is much better than the line. Further the residuals for the power law fit is distributed around zero with increasing scatter, which most likely is due to the increasing uncertainty that arise for enzymes with high K_m , where saturation is not possible. The residual for the linear regression show tendency to increase with K_m in a systematic manner, which indicates that the power law behavior (leveling off) is not captured by the line.

Regarding the application of the observed correlation between K_m and k_{cat} . We agree that this study do not include long-term effects, which will be relevant for industrial conversion of cellulose. Further studies are needed to understand how the scaling relations are effected by time/conversion one might also think about conversion as a drop in substrate load. Hence, for long-term reactions where the substrate is depleted, the optimal K_m will probably shift lower K_m values similar to what we illustrate for lower substrate loads in fig. 6. The optimal K_m in the initial phase (no/little substrate depletion) is shown to be $2.8 \cdot S_0$ but under high conversions S_0 will off course drop ($S(t) < S_0$ for $t > t_0$). We believe that the presented work may serve as a framework to understand and guide the design of new cellulases and that volcano plot like the ones shown in figure 6 may aid to rationalize comparative experimental data of different cellulases.

Reference

1. Sousa, S. F., Ramos, M. J., Lim, C., and Fernandes, P. A. (2015) Relationship between Enzyme/Substrate Properties and Enzyme Efficiency in Hydrolases. *ACS Catalysis* 5, 5877-5887

2. Warshel, A. (1998) Electrostatic Origin of the Catalytic Power of Enzymes and the Role of Preorganized Active Sites. *J. Biol. Chem.* **273**, 27035-27038
3. Sousa, S. F., Calixto, A. R., Ferreira, P., Ramos, M. J., Lim, C., and Fernandes, P. A. (2020) Activation Free Energy, Substrate Binding Free Energy, and Enzyme Efficiency Fall in a Very Narrow Range of Values for Most Enzymes. *ACS Catalysis* **10**, 8444-8453